

# The ESA GOME-Evolution "Climate" water vapor product: A homogenized time-series of H₂O columns from GOME/SCIAMACHY/GOME-2

Steffen Beirle [1], Johannes Lampel [1,*], Yang Wang [1], Kornelia Mies [1], Margherita Grossi [2], Diego Loyola [2], Angelika Dehn [3], Anja Danielczok [4], Marc Schröder [4], and Thomas Wagner [1]

[1]Max-Planck-Institut für Chemie (MPI-C), Mainz, Germany
[2]German Aerospace Centre (DLR), Remote Sensing Technology Institute, Oberpfaffenhofen, Germany
[3]European Space Agency (ESA), ESRIN, Frascati, Italy
[4]Deutscher Wetterdienst (DWD), Offenbach, Germany
[*]now at: Institut für Umweltphysik (IUP), Universität Heidelberg, Germany

*Correspondence to:* S. Beirle
steffen.beirle@mpic.de

**Abstract.** We present time-series of the global distribution of water vapor columns over more than two decades based on measurements from the satellite instruments GOME, SCIAMACHY, and GOME-2 in the red spectral range. Particular focus is the consistency amongst the different sensors to avoid jumps from one instrument to another. This is reached by applying robust and simple retrieval settings consistently. Potentially systematic effects due to differences in ground pixel size are avoided by

5      merging SCIAMACHY and GOME-2 observations to GOME spatial resolution, which also allows for a consistent treatment of cloud effects. In addition, the GOME-2 swath is reduced to that of GOME and SCIAMACHY to have consistent viewing geometries.

     Remaining systematic differences between the different sensors are investigated during overlap periods and are corrected for in the homogenized time series. The resulting "Climate" product (doi:10.1594/WDCC/GOME-EVL_water_vapor_climate)

10      allows to study the temporal evolution of water vapor over the last 20 years on global scale.



## 1 Introduction

Water vapor is a key component for the Earth's climate as it is an important natural greenhouse gas and it drives cloud formation. Thus, for reliable climate modeling, understanding the $H_2O$ cycle and possible feedback mechanisms is crucial. The analysis of the temporal evolution or trends of measured $H_2O$ on global scale is thus a key for improving our knowledge on the Earth's

climate system. International efforts are made to collect, improve and assess available water vapor measurements, e.g. within the GEWEX Water Vapor Assessment (http://gewex-vap.org), by the WMO World Climate Research programme.

Total column water vapor (TCWV) measurements can be made from radiosondes or from the analysis of GPS signals. Both techniques provide good coverage for e.g. North America and Europe where many ground stations exist, but only sparse coverage over e.g. Central Africa or the Oceans. Satellite measurements from microwave or infrared sensors, on the other hand,

are primarily sensitive over ocean/land, respectively.

Since the launch of GOME (see Table 1 for abbreviations and references) in 1995, spectral measurements of moderate resolution became available including the red spectral range, and have been continued by SCIAMACHY and GOME-2 up to now. These measurements allow the retrieval of TCWV (e.g., Noël et al. (1999); Wagner et al. (2003); Lang et al. (2003); Grossi et al. (2015)) by Differential Optical Absorption Spectroscopy (DOAS) (Platt and Stutz, 2008), providing global coverage with

similar sensitivity over both land and ocean. This TCWV data from satellite observations in the red spectral range has been used to investigate the water vapor evolution over time on global scale, e.g. the effects of El Niño (Wagner et al., 2005; Loyola et al., 2006) or trends (Wagner et al., 2006; Mieruch et al., 2008, 2011, 2014).

Table 1 about here.

The TCWV retrieval implemented in the operational GOME-2 data processor (GDP) (version 4.7) has been developed by

MPIC and DLR and is described in detail in Grossi et al. (2015). It is robust and almost independent of external data sets. Essentially, it is based on

(a) DOAS analysis, plus a simple correction for spectral saturation effects,

(b) empirical air-mass factors (AMFs) based on the $O_2$ absorption, and

(c) a simple cloud masking, again based on $O_2$ absorption.

These steps are shortly explained in section 2; for further details see Grossi et al. (2015) and references therein.

Within the ESA GOME-Evolution project, the "Climate" TCWV product has been developed. Goal of the Climate product is to provide an - as much as possible - consistent time-series of TCWV from GOME, SCIAMACHY, and GOME-2. This consistency is reached by spatial merging of SCIAMACHY and GOME-2 observations to GOME pixel size and by limiting the GOME-2 swath to that of GOME.

For the Climate product, the TCWV retrieval is at large parts similar to Grossi et al. (2015), i.e. requires almost no external data, but derives the information required for AMF correction and cloud masking directly from the spectral analysis. This allows for a consistent treatment of cloud effects for the different sensors, which would be difficult to achieve based on operational cloud products from different sensors and different algorithms.



The manuscript is organized as follows: In section 2, the TCWV retrieval used in the GDP is shortly summarized, and modifications for the climate product are explained. In section 3, the details of the spatial merging procedure are given (3.1), the consistency across the different instruments is checked during overlap periods (3.2), and homogenized time series are reached after offset correction (3.3). In section 4, some specific properties of the Climate product are discussed. Section 5 summarizes the results of validation studies. Details of and a link to the final data product are given in section 6, followed by conclusions in section 7.

## 2 TCWV retrieval

The retrieval of TCWV from satellite spectra for the climate product is based on the operational GDP TCWV retrieval described in Grossi et al. (2015). Below, we shortly summarize the single steps of the operational retrieval and point out where the climate algorithm differs. The particular operations for the climate product, i.e. the spatial resampling and the homogenization of time series across different satellite instruments, are described in the next section.

### 2.1 Spectral analysis

Slant column densities (SCDs), i.e. concentrations integrated along the effective light path, are derived from the satellite spectra using DOAS (Platt and Stutz, 2008). The retrieval is performed in the red spectral range from 614 to 683 nm, including the $O_2$ and $H_2O$ absorption bands at 630 and 650 nm, respectively. Within the spectral analysis, absorption spectra of $H_2O$, $O_2$, and $O_4$ are accounted for. In addition, an inverse irradiance spectrum and a "Ring"-spectrum are included, accounting for intensity offsets and Raman scattering, respectively. Furthermore, the spectral signatures from vegetation are considered by including the respective spectral structures deduced from deciduous, conifers, and grass absorption (Wagner et al., 2007). For SCIAMACHY, polarization correction spectra are included as well in order to account for its particularly strong polarization sensitivity. A polynomial of degree 4 is included in the fit.

Further details on and examples of the spectral retrieval can be found in Wagner and Mies (2011).

### 2.2 Saturation correction

Though atmospheric absorption depths of $H_2O$ and $O_2$ are only moderate in the considered wavelength range, still absorption can be considerable in the narrow absorption lines. For the measured spectra on moderate spectral resolution, absorption lines are smeared out by the instrumental spectral response function, resulting in a non-linear relationship between the actual TCWV and the retrieved $H_2O$ SCD. This effect can be simply modelled based on synthetic spectra as described in Wagner et al. (2003, 2006) for $H_2O$ and $O_2$, respectively. For the GDP and the climate retrieval, the $H_2O$ and $O_2$ SCDs resulting from the DOAS analysis are accordingly corrected for saturation effects.



## 2.3 Air Mass Factor

In passive DOAS applications, the derived SCD is usually converted into a vertical column density (VCD) by division with the so-called air-mass factor (AMF). The AMF depends on viewing geometry and the vertical concentration profile of the trace gas of interest, and is usually determined by RTM calculations. This is also the procedure used for the complementary GOME-

Evolution "Advanced AMF Algorithm ($A^3$)" product which is currently developed Wang et al. (2017). For the climate product, as for the GDP, however, we follow the approach proposed by Wagner et al. (2003) which takes the $O_2$ AMF as proxy for the $H_2O$ AMF. As the $O_2$ VCD is known, the $O_2$ SCD resulting from the DOAS fit (and corrected for saturation effects) directly yields the $O_2$ AMF. In order to account for the systematic difference in the vertical profiles of $O_2$ and $H_2O$, a correction factor depending on SZA and ground albedo is applied, which is determined from RTM calculations for standard atmosphere

conditions (see Grossi et al. (2015) for details). Note that, in contrast to the GDP, a scan angle dependent correction is not applied for the climate product for two reasons: 1. for the climate product, large scan angles ($>31°$), which occur for GOME-2, are skipped (see next section), and 2. the scan angle dependency is quite complex, i.e. different over land and ocean and depending on SZA as well, and the operational scan angle correction is still imperfect, as the resulting VCDs reveal remaining scan angle dependencies (Grossi et al., 2015).

The $H_2O$ VCDs (in units of molec/cm$^2$) directly correspond to TCWV (in units of kg/m$^2$). In the text hereafter, we use the term TCWV (except for issues directly related to the spectral analysis, i.e. SCDs). In the figures, both units (for VCD and TCWV) are given.

## 2.4 Cloud masking

For the climate product as for the GDP, a simple cloud masking is performed based on the retrieved $O_2$ SCD: As stated in

Wagner et al. (2006), pixels with less than 80% of the maximum $O_2$ SCD (as function of SZA) are masked as cloudy. Note that for the climate product, the maximum $O_2$ SCD was determined over the Pacific for each satellite instrument individually.

The drawbacks of this simple cloud masking are that low altitude clouds ($\lesssim$ 2-3 km) may be overlooked, and high mountains ($\gtrsim$ 2-3 km on GOME resolution) are misclassified as clouds (and thus skipped). The advantage of this approach (and the reason to stick to it for the climate product) is that is allows for a consistent cloud treatment across different satellite instruments, as

$O_2$ is derived simultaneously with $H_2O$ in the spectral analysis.

## 2.5 Gridding

The TCWV of the cloud-masked satellite pixels with SZA$<85°$ is gridded on a regular lat/lon grid with 1° resolution on daily basis. Backscans as well as the ascending part of the orbit are skipped. For GOME(-2), the narrow swath mode (NSM), which is applied about 3 (1) times a month, respectively, is discarded.

Subsequently, monthly means are calculated. Figure 1 exemplarily shows the monthly mean TCWV from GOME measurements in June 1996.

Figure 1 about here.



## 3 The climate product

The goal of the climate product is to provide an as best as possible consistent time series of TCWV from observations of the satellite instruments GOME, SCIAMACHY, and GOME-2, covering a time period of more than two decades. As indicated in table 1, the ground pixel size differs strongly between GOME and its successors. This has direct impact on the spatial resolution

of the resulting daily and monthly means, but in addition also more sophisticated consequences related to cloud masking, as the cloud statistics depend on pixel size (Krijger et al., 2007). Thus, for the climate product, "GOME-like" observations are generated from SCIAMACHY and GOME-2 by spatial resampling of SCIAMACHY and GOME-2 pixels to GOME size, and by reducing the GOME-2 swath to the swath of GOME and SCIAMACHY, as explained in detail in section 3.1. The consistency between GOME and the resampled SCIAMACHY and GOME-2 timeseries is checked in section 3.2. Finally,

homogenized time series are constructed by applying offset corrections to GOME and GOME-2 (section 3.3).

### 3.1 Spatial resampling to GOME pixel size and swath

The spatial resolution of GOME is considerably coarser than that of SCIAMACHY and GOME-2 (Table 1). Thus, in order to construct consistent time series amongst instruments, we have merged individual SCIAMACHY and GOME-2 observations down to GOME resolution.

The merging might be realized by co-adding the spectra of the respective satellite pixels. It is much easier, however, to use the existing $H_2O$ SCDs for SCIAMACHY and GOME-2, and determine the SCD of the merged pixels as radiance-weighted sum of the individual SCDs. We have checked this simplification and found very high correlation (R=0.99998) of the intensity weighted mean SCD with the "true" merged SCD based on co-added spectra. Slope and intercept of a linear fit are 1.0010 and 0.036 $kg/m^2$, respectively. Thus we followed this simplified approach. The $O_2$ SCDs, needed for AMF calculation and cloud

masking, are merged likewise. The SZA of the merged pixel (needed for the AMF correction factor) is calculated as the mean of all SZAs of the original pixels. Afterwards, the TCWV retrieval steps described above (sections 2.2-2.5) are performed for the spatially downsampled SCDs.

The GOME swath in nominal mode is 960 km wide, corresponding to a scan angle range of $\pm 31°$. The swath contains 3 "forescan" pixels of 320 km × 40 km (across × along track). Backscans as well as orbits with different scan patterns (like

NSM) are skipped for the climate product.

For SCIAMACHY, one scan consists of 16 forward pixels with 60 km width. These pixels can only approximately be merged into 3 GOME-like pixels. For sake of symmetry, we group the 5 westerly, 6 center, and 5 easterly pixels together, respectively. The grouping is based on the position of the scan mirror (ESM). Thereby, SCIAMACHY measurements with reduced integration time (corresponding to 30 km across track) are grouped consistently into 10 westerly, 12 center, and 10

easterly pixels. The small difference in along-track extent (30 km for SCIAMACHY vs. 40 km for GOME) cannot easily be accounted for and is ignored hereafter.

For GOME-2, grouping is done based on the scan mirror angle as well. Each 4 GOME-2 pixels are merged, matching exactly the extent of one GOME pixel. After 8 July 2013, when GOME-2 on Metop-A is switched to "narrow" mode (not to be mixed





with NSM; the narrow mode still covers half of the original GOME-2 swath, thus matching the GOME swath) in tandem operation with Metop-B, automatically 8 GOME-2 pixels (with 40 km width each) are merged by the scan angle selection. Pixels with scan angles >31° are skipped such that the swath width of the merged GOME-2 pixels matches that of GOME.

Note that for the illustration and discussion of the spatial resampling, results from SCIAMACHY and GOME-2 gained in
original resolution are indicated by the subscript "orig", while the respective reduced (with respect to spatial resolution and swath) product is indicated by the subscript "rdcd". Afterwards (from section 3.2.2 on), all SCIAMACHY and GOME-2 results are derived after spatial resampling on GOME resolution if not explicitly stated differently.

Figure 2 illustrates the original and merged ground pixels for SCIAMACHY compared to coincident GOME measurements (with a time difference of 29 minutes). The grouping of SCIAMACHY pixels is indicated by thick rectangles. Figure 2 clearly
illustrates the complex relation of spatial resolution and cloud masking, and suggests that the comparison between the merged SCIAMACHY pixels and GOME is far more meaningful than a comparison on original SCIAMACHY resolution. In the next section, it is shown that also on average the TCWV agrees much better between GOME and SCIAMACHY if the latter is spatially merged to GOME resolution.

Figure 2 about here.

**3.2   Comparison of different sensors**

In this section, TCWV from the different sensors are compared during the available overlap periods. We refer differences to SCIAMACHY, as it serves as link between GOME and GOME-2 timeseries. For the comparison between GOME and SCIAMACHY (section 3.2.1), the improved consistency gained by the adjustment of spatial resolution is clearly illustrated. The remaining systematic offsets between the different sensors are quantified. This will be used in section 3.3 for the composition
of a homogenized, cross-platform TCWV time-series.

**3.2.1   GOME versus SCIAMACHY**

Time series of global TCWV from GOME and SCIAMACHY overlap for the period August 2002 until June 2003. Afterwards, GOME has lost global coverage due the failure of the onboard tape recorder.

In Figure 3, we compare the mean difference between GOME and SCIAMACHY TCWV for the overlap period in three
different ways. Figure 3(a) shows the difference of the mean of monthly means, where SCIAMACHY data in original resolution is used. Here, for each dataset all available measurements are considered. In contrast, in Figure 3(b) the difference is determined from coincident measurements on orbital basis. This is possible as SCIAMACHY has the same orbital pattern as GOME with a time shift of half an hour. In Figure 3(c), the difference between GOME and coincident SCIAMACHY measurements with reduced resolution is shown.

30          Figure 3 about here.

The comparison of all available measurements for each instrument (Fig. 3(a)) shows large scatter, caused by the high variability of day-to-day atmospheric water vapor as well as clouds, and the different spatiotemporal sampling for both instruments



(missing orbits and SCIAMACHY gaps due to limb measurements). In contrast, the comparison of coincident measurements only (Fig. 3(b)) shows much smoother patterns, but now also clearly reveals systematic differences down to -3 kg/m$^2$ in the tropics. Note that this is of similar magnitude as the "level shifts" which have been applied in Mieruch et al. (2008) (see Fig. 13 therein) for the determination of trends from combined GOME/SCIAMACHY measurements.

The systematic difference is largely reduced when SCIAMACHY observations are resampled on GOME resolution (Fig. 3(c)). This is further illustrated in Fig. 4, where zonal means of GOME and SCIAMACHY TCWV and their difference are shown as function of latitude. Over ocean, the resampled SCIAMACHY TCWV agrees with GOME within ±0.5 kg/m$^2$, whereas the original SCIAMACHY TCWV is systematically higher by about 0.3 kg/m$^2$ for mid and high latitudes, up to 1.0 kg/m$^2$ around the equator. Over land, good agreement is found between GOME and SCIAMACHY except for the tropics. Here, the merging
of SCIAMACHY pixels halves the systematic difference from -1.0 kg/m$^2$ down to -0.5 kg/m$^2$.

Figure 4 about here.

### 3.2.2 GOME-2 versus SCIAMACHY

Between GOME-2 and SCIAMACHY, a far longer overlap period is available (January 2007 until March 2012). However, in contrast to the comparison between GOME and SCIAMACHY, the selection of coincident measurements is not beneficial,
since the orbital patterns of GOME-2 and SCIAMACHY are shifted in longitude with respect to each other, and the swath width of GOME-2 has been reduced for the merged pixels. Thus, "coincident" measurements (with respect to time) are only available for a subset of the orbit, with systematic differences of the respective scan angles of the two instruments.

Thus, the mean difference of TCWV from GOME-2 and SCIAMACHY is calculated as mean of monthly means of all available measurements (Fig. 5). Though more than 5 years of data, the resulting difference is still noisy, due to the high
spatio-temporal variability of H$_2$O and clouds. In addition, it still reveals systematic orbital patterns, in particular over ocean. These, however, are not caused by individual orbits, but turned out to be a consequence of the GOME-2 NSM, which is periodically applied at the same geolocations, as demonstrated and discussed in detail in Appendix B.

Over land, GOME-2 TCWV is higher than SCIAMACHY by up to 2 kg/m$^2$ locally over tropical rain forest. In the zonal mean, GOME-2 and SCIAMACHY agree within ±0.3 kg/m$^2$. Over ocean, the zonal mean difference is again close to zero at
high latitudes, but goes down to about -1 kg/m$^2$ at the equator.

Figure 5 about here.

Figure 6 about here.

### 3.2.3 GOME-2 versus GOME

GOME has lost global coverage due to failure of the onboard tape recorder in June 2003, but continues measurements until
July 2011. During that period, the measured spectra have been directly transmitted to an increasing number of ground stations, mostly in the Northern hemisphere. This allows us to also directly compare GOME-2 and GOME at least for selected regions.





Like for the comparison between GOME-2 and SCIAMACHY, coincidence is not demanded. Since the results over ocean are quite noisy again, we perform the comparison separately over land and ocean.

Figure 7(a) displays the mean difference of GOME-2 and GOME TCWV over land for regions with sufficient coverage. In Figure 7(b), we also derived an indirect comparison between GOME-2 and GOME via the respective differences to SCIA-
MACHY, i.e. (GOME-2-SCIAMACHY)-(GOME-SCIAMACHY), for the same regional selection.

Figure 8 displays zonal mean difference between GOME-2 and GOME over ocean, again determined both directly and indirectly.

Figure 7 about here.

Figure 8 about here.

Thus, though GOME has lost global coverage in June 2003, the ongoing measurements still serve as valuable consistency check and reveal that a direct comparison to GOME-2 yields basically the same results as the two-step comparison via SCIA-MACHY. But due to the low spatial coverage, which is also changing over time, GOME measurements after June 2003 are not included in the merged time series.

### 3.3   Merged time series

As shown in the previous section, the resampling of SCIAMACHY and GOME-2 pixels to GOME resolution and swath width substantially improves consistency across the different instruments. But still, the comparison of mean TCWV during overlap periods reveal systematic regional differences between the different instruments, in particular in the tropics. These differences might be caused by instrument characteristics (like polarisation sensitivity or spectral resolution) and spatial/temporal sampling effects (Coldewey-Egbers et al., 2015). In addition, the different local overpass times (table 1) might cause systematic
differences in case of diurnal cycles of TCWV or cloud conditions. As shown in Diedrich et al. (2016), the change of TCWV between 9:30 and 10:30 local time is typically low ($<1\%$, which still might account for some tenths of kg/m$^2$ in the tropics). Systematic effects caused by change of cloud fraction and height are complex as they affect both the cloud masking (sect. 2.4) and the AMF (sect. 2.3). In particular over dark surfaces like the tropical rain forest, this can have significant impact, and is probably the reason for the offsets between GOME(-2) and SCIAMACHY over Brasil, which have opposite signs for GOME
and GOME-2.

If such systematic differences between the instruments would not be accounted for in the TCWV timeseries, discontinuities ("jumps") would occur (compare Mieruch et al. (2008)) which impair the analysis of trends if not considered. For the climate product, the timeseries from GOME, SCIAMACHY and GOME-2 are thus homogenized by applying offset corrections derived from the overlap periods. GOME and GOME-2 are corrected with respect to SCIAMACHY, as the latter serves as link between
GOME and GOME-2 timeseries.

GOME is corrected by subtracting the offset derived during the overlap with SCIAMACHY (Fig. 3(c)) after applying slight spatial smoothing (see Appendix A for details). For GOME-2, the offset (Fig. 5) is smoothed likewise over land; over ocean,





however, the slight smoothing is not sufficient to overcome the patchyness of the observed difference. Thus, the zonal mean TCWV is taken for all longitudes over ocean . The resulting offset corrections are displayed in Fig. 9.

Figure 9 about here.

The climate product provides a merged time series covering the period July 1995 until December 2015. Herein, GOME and GOME-2 monthly means are corrected with respect to the offset determined from comparison to SCIAMACHY. During overlap, measurements from all available instruments are averaged. Due to the higher spatial coverage of GOME and GOME-2 compared to SCIAMACHY, the monthly means are dominated by GOME(-2) measurements if available.

## 4 Known issues

The climate product is optimized for consistent time series across different satellite instruments. It is thus based on a simple retrieval, merged pixels, and reduced swath of GOME-2, at the cost of algorithm accuracy, spatial resolution, and spatial coverage. Below we list some aspects of the climate product that have to be kept in mind for data interpretation and comparison to other TCWV products.

### 4.1 Spatial resolution

GOME has a coarse across-track resolution of 320 km. For the climate product, also SCIAMACHY and GOME-2 observations are merged to GOME resolution. Thus, gradients in TCWV or in quantities affecting the AMF (like surface albedo, terrain height, or clouds), are not resolved but smeared out in the climate product. Systematic biases of the climate product TCWV are thus expected e.g. for coastal sites, and in particular for mountainous islands (compare Van Malderen et al., 2014).

### 4.2 Spatial gaps

The cloud flagging based on $O_2$ SCDs (sect. 2.4) discards observations over high mountains, resulting in persistent gaps over the Himalayas, the Andes, or Antarctica.

In individual monthly means, additional gaps occur if no "cloud free" measurements were found within an $1° \times 1°$ pixel. This regularly happens, mostly around the ITCZ, in particular for SCIAMACHY due to the poorer spatial coverage resulting from the alternating nadir/limb mode.

An additional gap is introduced by GOME calibration measurements which are regularly performed North from India.

### 4.3 Accuracy

The TCWV Climate algorithm applies a simple empirical AMF correction based on the observed $O_2$ SCDs. The impact of the different vertical profiles of $H_2O$ and $O_2$ is corrected for based on mean $H_2O$ profiles determined from an average lapse rate. For individual observations, actual AMFs might deviate considerably if the $H_2O$ profiles differ from the mean, especially if



clouds are present. This might also affect monthly means in case of systematic differences. However, the simple and robust settings allow for a consistent retrieval (including the treatment of clouds) across the different instruments.

Comparisons to independent measurements result in relative biases of typically -5% to -10% for the total mean (see section 5).

## 4.4 Scan angle dependency

As documented in (Grossi et al., 2015), the GDP TCWV retrieval requires empirical corrections of systematic scan angle dependencies of both $H_2O$ and $O_2$ SCDs, which are particularly strong over ocean. In the climate product, no corrections of scan angle dependencies are applied, as the large viewing angles of GOME-2 are skipped by reducing the swath with to that of GOME. In addition, the scan angle dependencies also depend on further quantities like SZA and surface albedo, and are thus hard to correct for appropriately (see (Grossi et al., 2015) for detailed discussion).

However, the effects of scan angle dependencies on TCWV are usually reduced in monthly means, and cancel out in longer temporal averages, as long as the spatial sampling with different scan angles is uniform for the different sensors. This is usually the case, as shown in Appendix B, with two prominent exceptions:

- For GOME, systematic scan angle biases occur around the calibration region over Northern India, as locally measurements from the Eastern or Western swath pixels dominate.

- For GOME-2, the narrow swath mode (NSM) is applied regularly at the same geolocations. As the NSM is skipped in the Climate product, these regular gaps result in biased mean scan angles with systematic orbital patterns. This is the reason for the small but systematic orbital patterns in the mean difference between GOME-2 and SCIAMACHY TCWV during overlap period (Fig. 5). For the applied offset correction, these patterns are removed by taking the zonal mean over ocean for all longitudes (Fig. 9(b)).

In the climate product, a "warning flag" is provided indicating regions where the mean scan angle systematically deviates from 0. In addition, the mean scan angles for each instrument as shown in Fig. A1 are provided so that the user might check whether suspicious spatial patterns might be related to a scan angle bias.

In order to avoid orbital artefacts caused by systematic scan angle biases in the climate product, a second version of the climate TCWV timeseries ("TCWV$_{smooth}$") is added to the data product where monthly means are smoothed over ocean such that the orbital patterns are removed. Details of the applied smoothing are provided in Appendix A. The original and smoothed monthly mean TCWV are shown in Fig. 10 for September 2015 exemplarily.

Figure 10 about here.

## 5 Validation

Within the ESA GOME-Evolution project, the Climate product has been validated by comparison to TCWV retrieved from Global Navigation Satellite System measurements (GNSS, Wang et al. (2007), version 721.1) as well as from the Analysed





RadioSoundings Archive (ARSA, version 2.7). Below we shortly summarize the validation results and refer to the validation report (Danielczok and Schröder, 2017) for further details. Note that

- the validation report is based on the climate product v2.01. The current version 2.1 presented here is using exactly the same TCWV algorithm, but a slightly different definition of the warning flag. In addition, June 1995 was included in v2.01, but skipped in v2.1 due to the limited number of available GOME measurements, resulting in a noisy monthly mean.

- validation is based the TCWV data product. Validation results for $\text{TCWV}_{\text{smooth}}$ are basically the same, since the smoothing is only applied over ocean. With GNSS/ARSA data, an appropriate validation of $\text{TCWV}_{\text{smooth}}$ is thus not possible.

.

## 5.1 Accuracy

Figures 11 and 12 display scatter plots of monthly mean TCWV from GNSS and ARSA stations, respectively, compared to the climate product. TCWV from GNSS and ARSA show good correlation to the climate product. Mean biases are -1.0 kg/m$^2$ and -1.9 kg/m$^2$, respectively. If only station measurements around the satellite local overpass time are considered, biases are reduced to -0.7 kg/m$^2$ and 0.2 kg/m$^2$, respectively. The respective RMS for the comparison of the climate product to GNSS/ARSA are 4.4/5.6 kg/m$^2$ (all measurements) and 4.3/6.1 kg/m$^2$ (around 10:00 local time), respectively (Danielczok and Schröder, 2017).

On smaller spatio-temporal scales (seasonal, regional), biases can be higher (several kg/m$^2$) and can even exceed $\pm 15$ kg/m$^2$ at low latitudes, in particular for coastal sites (probably related to the coarse spatial resolution of the climate product). Further comparisons to TCWV from ECMWF and HOAPS reveal biases of the same order of magnitude Grossi (2017).

Overall, the observed biases are comparable to those that have been reported for the GOME-2 GDP 4.7 product in (Grossi et al., 2015) and can be understood by the simplifications made in the climate product retrieval (compare section 4.3).

Figure 11 about here.

Figure 12 about here.

## 5.2 Temporal stability

As the focus of the climate product is to provide stable TCWV time-series, particular validation focus is put on the temporal stability of the TCWV product. Figure 13 displays the relative difference between monthly mean TCWV from Climate product and GNSS for stations available over the whole time range as function of time. The time series shows a small, but significant trend of -1.7% per decade. A similar comparison with ARSA data, however, reveals a small significant positive trend of 0.5% per decade (Fig. 14). The different results are probably caused by the different spatial distribution of stations, and sampling effects of the Climate product (a local "monthly mean" is often determined from less than 5 satellite measurements). If GNSS and ARSA measurements are selected close to the time of satellite overpass around 10:00 local time, slopes are -0.91 and





-0.89% per decade, respectively, and both significantly different from 0% per decade (Danielczok and Schröder, 2017). Note that most of the ARSA data is provided for 0:00 and 12:00 UTC. Thus, the selection of 10:00 local time reduces the available ARSA stations to 1/5.

Figure 13 about here.

5                                                                                       Figure 14 about here.

The stability of the Climate product is thus approximately $\pm 1\%$ per decade, which is stated as requirement for infering trends in TCWV in (Saunders et al., 2010). However, the spatial coverage of ARSA and GNSS stations is rather poor in some regions and the majority of the open oceans are not covered at all. As the stability can be a strong function of region Schröder et al. (2016), future efforts are needed to assess stability globally.

Note that the transitions between the satellite instruments are not visible in the timeseries of the difference to the validation data sets.

## 6   Data format and availability

The GOME-Evolution Climate product is available at World Data Center for Climate (WDCC):

doi:10.1594/WDCC/GOME-EVL_water_vapor_climate

The data is provided as single netCDF4 file, containing monthly mean TCWV with 1° resolution. The period from July 1995 to December 2015 is covered. Dimensions are

- time: months since 1995 (starting with 7 corresponding to July 1995)
- latitude: latitude of grid pixel center (89.5°N to 89.5°S)
- longitude: longitude of grid pixel center (179.5°W to 179.5°E).

Two TCWV fields are provided:

- TCWV(time, lat, lon): For each month, global maps of TCWV are provided with 1° resolution based on the (offset corrected) monthly means from GOME, SCIAMACHY, and GOME-2.
- TCWV_smooth(time, lat, lon): additional spatial smoothing of monthly means is applied over ocean, see section 4.4).

Unit is kg/m$^2$.

In addition, the following data fields are provided:

- Contribution_from_xy[1](time): flag indicating to which months the respective instrument is contributing.
- Mean_Scan_Angle_xy(lat, lon): maps providing the mean scan angle. See Appendix B.
- Warning_Flag(lat, lon): indicates large positive/negative mean scan angles for at least one instrument.

---

[1]xy = GOME1/SCIA/GOME2





# 7 Conclusions

The GOME-Evolution Climate water vapor product provides a consistent global time series of TCWV derived from the satellite instruments GOME, SCIAMACHY, and GOME-2 (MetopA) covering two decades. Consistency is reached by merging SCIA-MACHY and GOME-2 observations to GOME pixel size and reducing the GOME-2 swath width to GOME / SCIAMACHY swath. Part of the remaining differences between the instruments are due to instrument characteristics and the different local overpass time, which might be relevant in case of systematic diurnal cycles of $H_2O$ or in particular clouds from 9:30 to 10:30 local time. The time series are homogenized by determining the offsets during overlap periods and correcting for them, resulting in temporal stability of about 1% per decade as demonstrated by comparison to independent TCWV datasets.

*Acknowledgements.* The generation and validation of the Climate product was funded by the ESA "GOME-Evolution" project under contract number 4000110429.

Spectral measurements from GOME and SCIAMACHY are provided by ESA. Spectral measurements from GOME-2 are provided by EUMETSAT.

The ARA/ABC(t)/LMD group and as well as NCAR/UCAR/EOL are acknowledged for producing and making available the ARSA and GNSS data, respectively.





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

**Table 1.** Characteristics of the satellite instruments used in this study

|  | GOME | SCIAMACHY | GOME-2 |
|---|---|---|---|
| Instrument | Global Ozone Monitoring Experiment | SCanning Imaging Absorption spectroMeter for Atmospheric CHartographY | Global Ozone Monitoring Experiment 2 |
| Satellite | ERS-2 | ENVISAT | Metop-A[a] |
| Launch | 1995 | 2002 | 2006 |
| Temporal coverage[b] | July 1995 - June 2003 July 2003 - July 2011 (reduced spatial coverage) | August 2002 - March 2012 | January 2007 - [c] |
| Regular footprint [km$^2$] | $40 \times 320$ | $30 \times 60$ | $40 \times 80$ ($40 \times 40$[d]) |
| Swath width [km] | 960 | 960 | 1920 (960[d]) |
| Local time | 10:30 | 10:00 | 9:30 |
| Reference | Burrows et al. (1999) | Bovensmann et al. (1999) | Munro et al. (2016) |

[a] A second GOME-2 instrument was launched 2012 on Metop-B, and a third is planned to be launched on Metop-C in 2018.

[b] i.e., TCWV available.

[c] Within the climate product v1.0, GOME-2 data until December 2015 is included.

[d] Since July 2013 in Metop-A/B tandem operation.



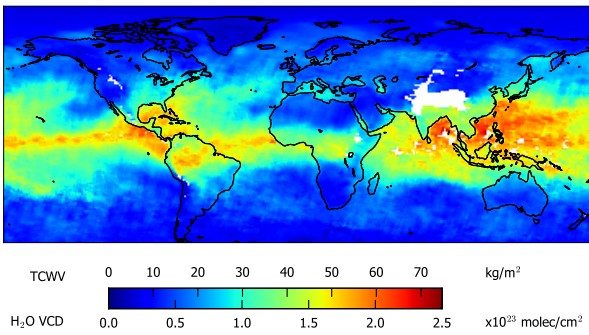

**Figure 1.** Sample monthly mean TCWV from GOME measurements in June 1996 .

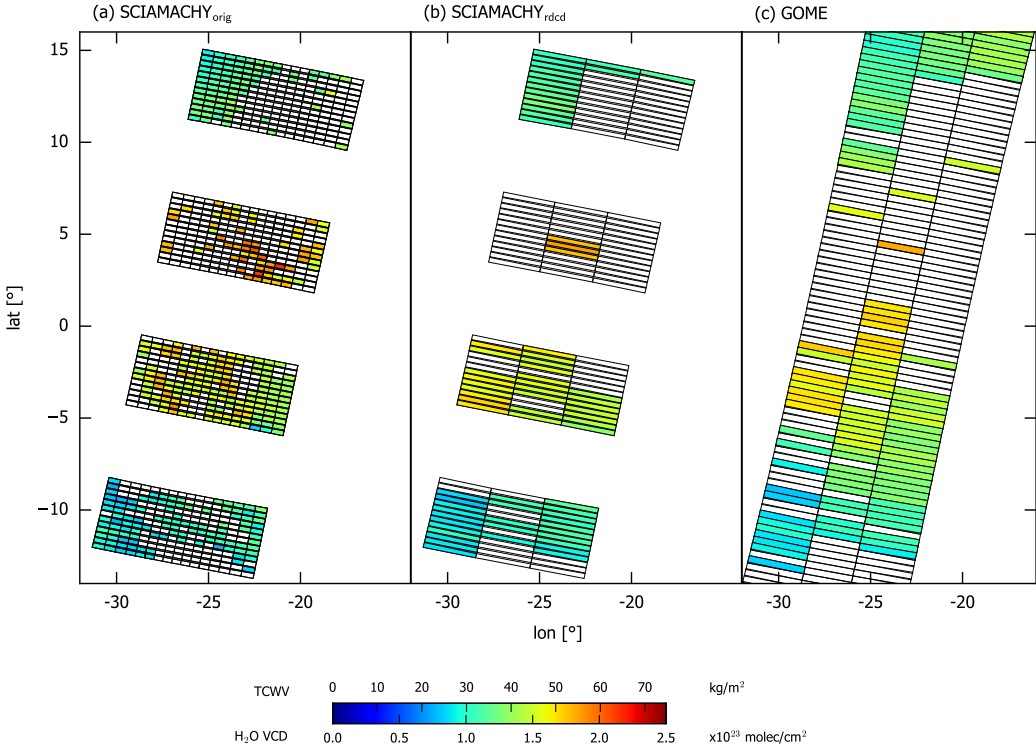

**Figure 2.** TCWV from SCIAMACHY in original (a) and reduced (b) resolution, compared to GOME (c), on 1st of June 2003. White pixels are masked by the cloud flag as described in section 2.4. Spatial gaps in the SCIAMACHY orbit are caused by switches to limb observations. Time shift between SCIAMACHY and GOME is 29 minutes.


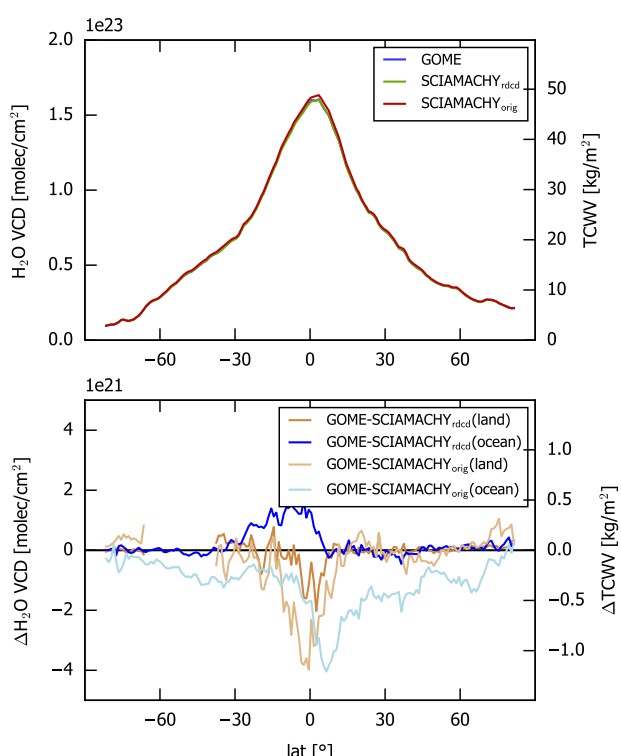

**Figure 4.** Zonal mean TCWV for GOME and SCIAMACHY (top) and the respective differences, separately for land and ocean (bottom), as function of latitude.

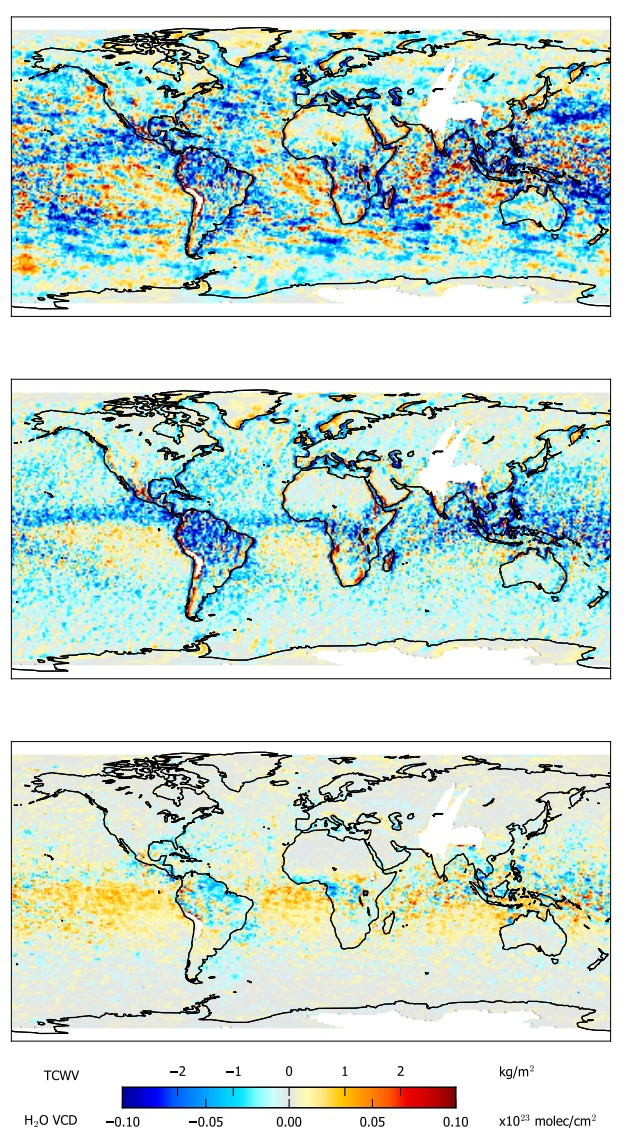

**Figure 3.** Mean difference of GOME and SCIAMACHY TCWV during the overlap period August 2002 to June 2003 calculated as mean of monthly means (a) or as mean of coincident measurements on orbital basis (b), (c). In (a) and (b), SCIAMACHY data is taken in original resolution. In (c), SCIAMACHY pixels are merged to GOME resolution.

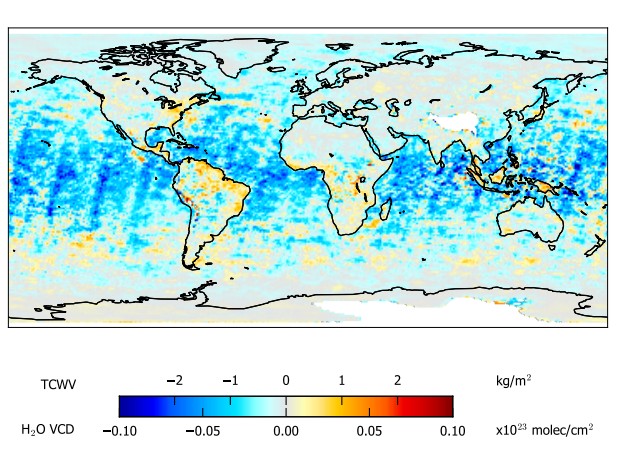

**Figure 5.** Mean difference of GOME-2 and SCIAMACHY TCWV during the overlap period January 2007 to March 2012 calculated as mean of monthly means.

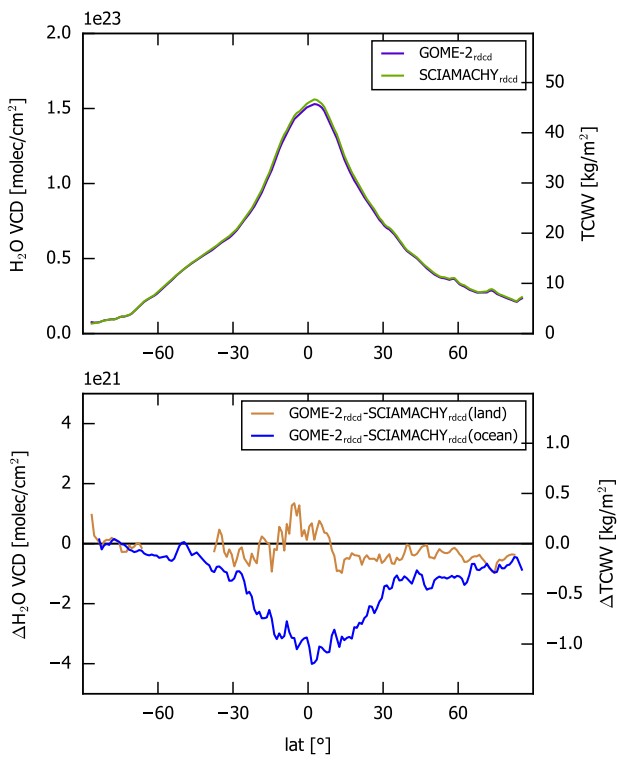

**Figure 6.** Zonal mean of TCWV for GOME-2 and SCIAMACHY (top) and the respective differences, separately for land and ocean (bottom), as function of latitude.

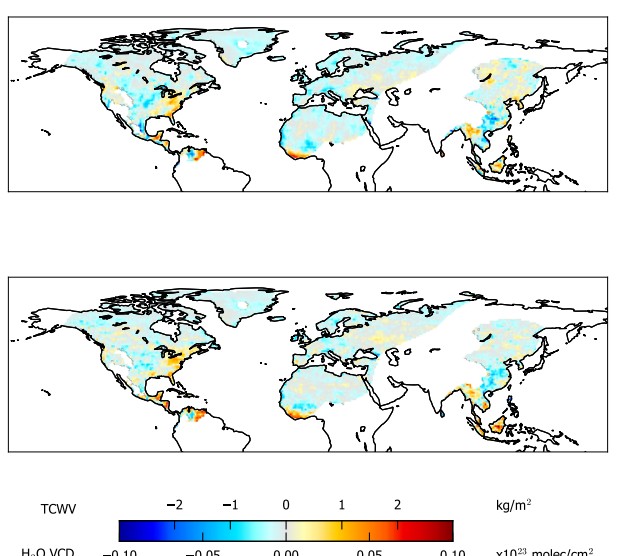

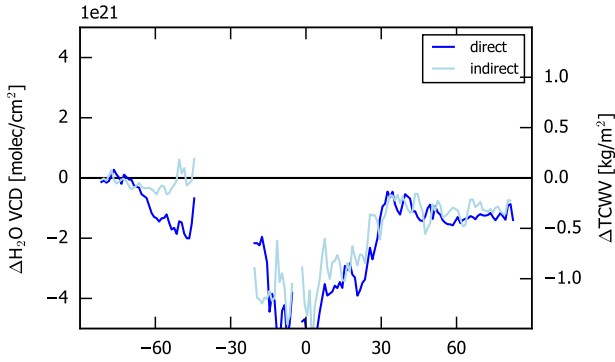

**Figure 7.** Mean difference of TCWV between GOME-2 and GOME after tape recorder failure during the overlap period January 2007 to February 2010 calculated as mean of monthly means (a). Oceans and regions with poor GOME coverage are masked out. For comparison, (b) displays the indirect difference between GOME-2 and GOME, as derived from the difference between GOME-2 and SCIAMACHY (Fig. 3c) minus the difference between GOME and SCIAMACHY (Fig. 5) for the same spatial selection.

**Figure 8.** Zonal mean of direct and indirect TCWV differences between GOME-2 and GOME over ocean as function of latitude.

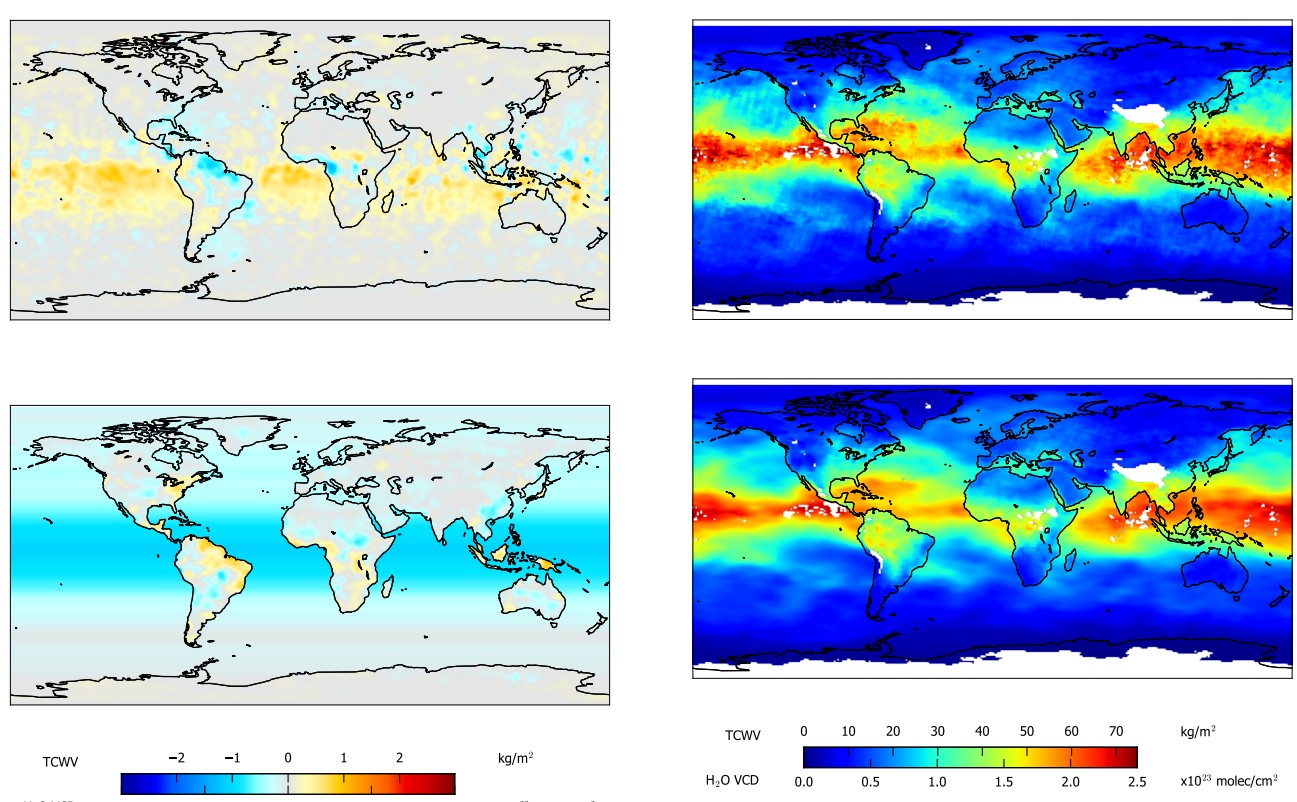

**Figure 9.** Final offset correction applied to GOME (a) and GOME-2 (b). See text for details.

**Figure 10.** Monthly mean TCWV (top) versus TCWV$_{\mathrm{smoothed}}$ (bottom) for September 2015.



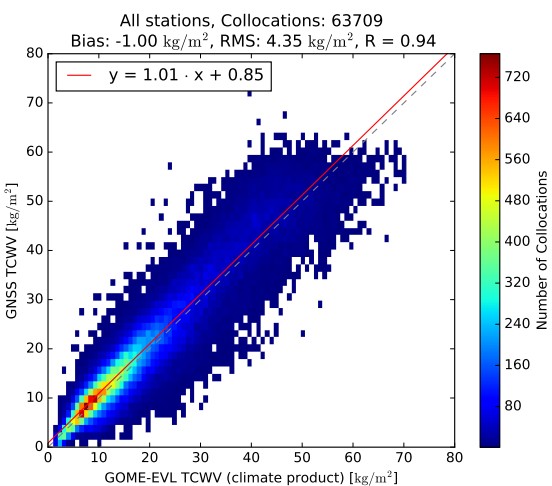

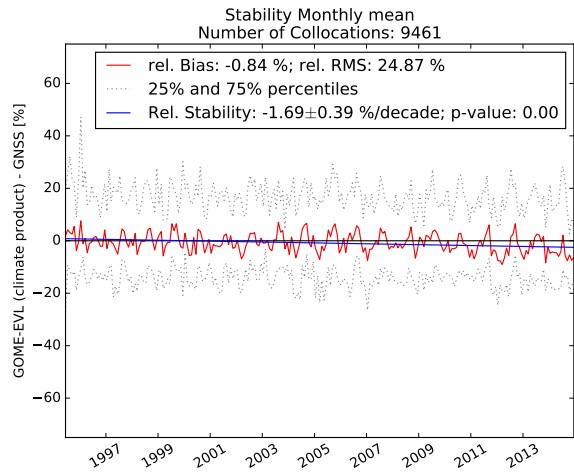

**Figure 11.** Scatter plot of TCWV monthly means of all available GNSS stations and the climate product. Figure from (Danielczok and Schröder, 2017).

**Figure 13.** Timeseries of the relative difference between TCWV monthly means from the Climate product and GNSS stations. Only stations which were available for the whole time period are considered. Figure from (Danielczok and Schröder, 2017).

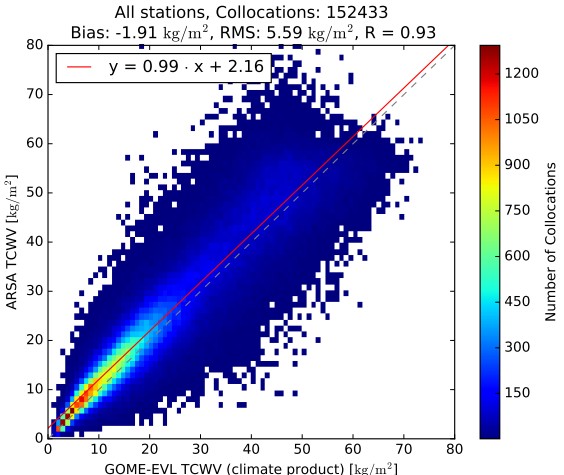

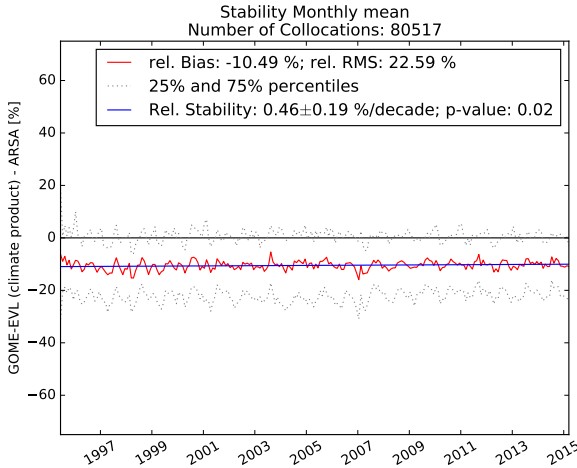

**Figure 12.** Scatter plot of TCWV monthly means of all available ARSA stations and the climate product. Figure from (Danielczok and Schröder, 2017).

**Figure 14.** Timeseries of the relative difference between TCWV monthly means from the Climate product and ARSA stations. Only stations which were available for the whole time period are considered.



## Appendix A: Convolution kernels for spatial smoothing

Spatial smoothing is realized as normalized convolution (Knutsson and Westin, 1993) of monthly mean TCWV maps with a convolution kernel (CK) $C$ on a regular 1° latitude/longitude grid. In contrast to basic matrix convolution, normalized convolution can be applied to matrices containing gaps and removes them. Below we provide the CKs used for the smoothing of

5  offset maps (sect. 3.3) and for the smoothed climate product (sect. 4.4).

### A1  Smoothing of offsets

For the smoothing of offsets (sect. 3.3), the CK

$$C_{\text{offset}} = \begin{bmatrix} 1 & 1 & 1 & 1 & 1 & 1 & 1 \\ 1 & 3 & 3 & 3 & 3 & 3 & 1 \\ 1 & 3 & 10 & 10 & 10 & 3 & 1 \\ 1 & 3 & 10 & 10 & 10 & 3 & 1 \\ 1 & 3 & 10 & 10 & 10 & 3 & 1 \\ 1 & 3 & 3 & 3 & 3 & 3 & 1 \\ 1 & 1 & 1 & 1 & 1 & 1 & 1 \end{bmatrix} /162$$

is used, which is basically a 3x3 boxcar, enlarged to 7x7 with lower elements at the edges. This removes gaps from the offset

10  maps, while local structures are preserved (compare Figures 3(c) and 9(a)).

### A2  Smoothing of climate product

For the smoothed climate product, smoothing is applied primarily zonally in order to remove the artificial orbital patterns over ocean. For this task, the CK

$$C_{\text{smooth}} = \begin{bmatrix} 1 & 1 & 1 & 1 & 1 & 1 & 1 & 1 & 1 & 1 & 1 \\ 1 & 1 & 2 & 3 & 3 & 3 & 3 & 3 & 2 & 1 & 1 \\ 1 & 1 & 1 & 1 & 1 & 1 & 1 & 1 & 1 & 1 & 1 \end{bmatrix} /45$$

15  is applied, which is 11° wide. $C_{\text{smooth}}$ is only applied over ocean. Its impact is illustrated in Fig. 10.

Note that the convolution with $C_{\text{smooth}}$ is not used to fill gaps in order to avoid data entries at locations where actually no measurements are available. I.e., after normalized convolution, any originally missing value in TCWV is removed from TCWV_smooth as well.



**Appendix B: Mean scan angles**

The retrieved TCWV of individual observations shows a scan angle dependency (SAD), in particular over ocean, resulting from the scan angle dependencies of both $O_2$ and $H_2O$ SCDs (see 4.4). In the climate product, this is not explicitly corrected. However, SAD effects are reduced in monthly means, and cancel out in longer temporal averages, as long as the mean scan
angle is close to 0 (=nadir). Systematic biases of the mean scan angle, on the other hand, can cause small but systematic biases of the mean TCWV.

Fig. A1 displays the mean scan angle (mean of monthly means) for the considered sensors which are discussed below.

**B1  GOME**

For GOME, the mean scan angle is generally close to 0. But around the calibration region over Northern India, large systematic
biases are observed, as locally measurements from the Eastern or Western swath pixels dominate. Less pronounced scan angle biases are observed for orbital fragments south from India and around 140°-160° E.

**B2  SCIAMACHY**

For SCIAMACHY, the mean scan angle is close to 0 all over the world. A calibration gap as for GOME does not exist. Note that the overall average is slightly negative. This is caused by an asymmetry of the SCIAMACHY scan pattern ranging from
-31 to +29 ° (see table 3-3 in Gottwald et al. (2010)). This is accounted for in the spatial merging of SCIAMACHY pixels to GOME resolution by adjusting the scan angle thresholds. Consequently, any (small) bias potentially caused by the systematic negative SCIAMACHY scan angles is corrected for by the offsets determined during overlap periods.

**B3  GOME-2**

GOME-2 performs measurements in NSM periodically at the same geolocations (GOME-2 Factsheet, 2015). As NSM orbits
do not cover the full GOME swath, they are skipped in the climate product. This results in orbital patterns of inhomogeneous sampling with respect to scan angles.

The clear orbital patterns reflect the locations of the NSM orbits, as specified in table 7 in GOME-2 Factsheet (2015). Note that over Europe/West Africa/South Atlantic, the NSM is not applied regularly, resulting in mean scan angles close to 0. Over the US and Western Pacific, the deviations from 0 are particularly large due to additional orbits in nadir static mode, which
are skipped as well in the climate product. The large values East from Japan are caused by the fact that NSM is applied to 15 orbits; the first and the last of these orbits are close to each other.

Due to the SAD of the retrieved TCWV for individual observations, these orbital patterns are reflected in the mean difference between SCIACHY and GOME-2 (Fig. 5). For this reason, the offset GOME-2 minus SCIAMACHY over ocean is derived from the respective zonal mean (see section 3.2.2).



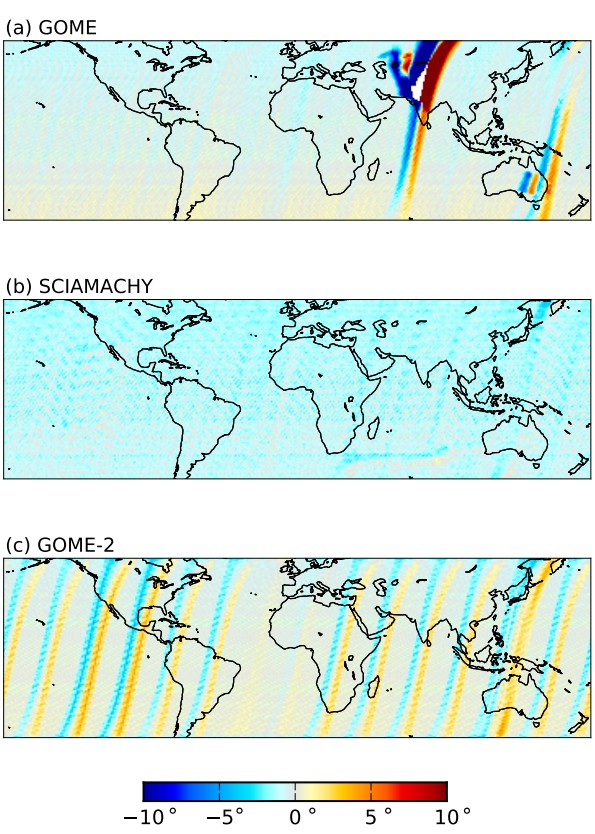

**Figure A1.** Mean scan angle for GOME (a), SCIAMACHY (b), and GOME-2 (c). Backscans as well as orbits in NSM have been skipped. See text for details.