# Peer review of "The ESA GOME-Evolution "Climate" water vapor product: A homogenized time-series of H2O columns from GOME/SCIAMACHY/GOME-2"

_Earth System Science Data, 2017_

## Referee Comment (RC1) · Anonymous Referee #1 · 12 Oct 2017

**General Comments:**

The paper by Beirle et al. describes a specific water vapour data set, which is based on a combination of existing water vapour products for GOME, SCIAMACHY and GOME-2.

Although the underlying data and methods are not new, the resulting combined and homogenised data set may be useful for future climate studies. The data set is available via the specified link, but only to registered users; therefore download (and the data

product itself) could not be checked.

The paper is well written and contains (except for the points listed below) all required information for a potential data users, including information about the quality of the data set. However, it is not clear how some of the references especially in the validation section can be accessed (see below).

My main points of criticism are:

1. Scan angle correction:
   It is stated in Section 2.3 that this is not applied for the climate data product because of the smaller scan angle range and the complexity/quality of the correction. However, as shown in Fig. A1, there are especially for GOME-2 significant scan angle effects. Although the scan angle related patterns are (at least partly) removed for the offset correction, they are still left in the climate product (at least in the unsmoothed one). The product contains a corresponding warning flag, but from Fig. A1 it seems that a lot of data will be affected. The impact of not applying a scan angle correction on the data product should be quantified and a clear recommendation should be given to the data users, if the flagged data should be used or not (or, under which conditions).

2. Product errors:
   The climate data product does not contain any TCVW errors. There is no description on how errors of the underlying products are considered. If it is not possible to specify an error for each data point, at least some general information about the expected quality (independent from validation results) should be given. For example, I would expect different quality/errors for the different instruments (and therefore for different time intervals), simply because of the merging of SCIAMACHY and GOME-2 pixels to GOME spatial resolution.
**Specific Comments:**

1. p. 4, l. 7–10:
   Using the $O_2$ AMF as proxy for $H_2O$ AMF assumes that the $O_2$ VCD is known. What is usually known is the $O_2$ VMR, but the VCD should also depend on (varying) pressure and temperature. The additional correction factor for different profile shapes is only determined from standard atmosphere conditions. Is there a remaining dependency on pressure and temperature? Since this applies to both $O_2$ and $H_2O$ in a similar way the final impact might be low, but maybe this should be mentioned in the text.

2. p. 4, l. 22–23:
   Where does the range 2–3 km come from – is this related to the 80% $O_2$ SCD threshold? I assume this should depend also on cloud fraction?

3. p. 5, l. 30–31:
   The along track spatial resolution is given by the product of the along-track velocity of the satellite and the measurement (integration) time (plus along track size of the field of view). Since orbital parameters and along track field of view sizes of GOME and SCIAMACHY are very similar, the main difference is the integration time. Merging the SCIAMACHY data such that the across track spatial resolution matches the one of GOME should therefore result in a quite similar along track spatial resolution. So, actually the merging accounts for the difference in along track extent, which is then even smaller than the mentioned 30 km vs. 40 km.

4. p. 7, l. 10:
   Is there a reason for the larger differences in the tropics? Could this be related to the above mentioned issue that corrections for the different profile shapes are based on average atmospheric conditions or maybe the probably in the tropics larger (and possibly less accurate) saturation correction? Please discuss.
5. p. 10, l. 24–26:
   Why is the smoothing only applied to ocean data? The scan angle dependent artefacts also occur over land (see Fig. A1).

6. Section 5:
   The section on validation mainly refers to a report by Danielczok and Schröder (2017). Where is this report available? If it is not available to the public a bit more information about the validation activities should be given (collocation criteria, averaging procedures etc.).

   Furthermore, from the description it is not clear where the GNSS/ARSA stations are located – I suggest to add a corresponding map. Probably these stations are mostly at land, so how good is the information about the quality of the data product over ocean?

7. p. 11 , l. 18:
   Where is the reference by Grossi(2017) available?

8. p. 12 , l. 10–11:
   *'Note that the transitions between the satellite instruments are not visible in the timeseries of the difference to the validation data sets.'*

   Is this because of the averaging of data during the overlap times?

9. p. 12, l. 15:
   It should be mentioned earlier (e.g. in the introduction) that the climate product contains monthly mean data at 1 deg resolution.

10. p. 12, l. 28:
    How many data are affected by the Warning_Flag? Looking at Fig. A1 and noting that there is no distinction between time and/or instrument this should be quite a lot. I suggest to add a plot of the Warning_Flag to Fig. A1 and an average number of affected data points to have a better impression on this.

11. Fig. 2:
It seems that the GOME ground pixels across track have different size (centre pixel is smaller) whereas coadded SCIAMACHY pixels have about the same size. Could this have been improved by different coadding patterns for SCIAMACHY?

12. Figs. 13 & 14:
Fig. 13 shows a small decrease of the differences towards the end of the time series (probably the reason for the negative stability). This decrease is not seen in Fig. 14, which indicates that it is not related to the satellite data (e.g. due to GOME-2 degradation). Is there any information about the quality of the station data?

13. Appendix A:
A bit more information should be given about the 'normalised convolution' and how it is exactly applied. What are the limitations for removing gaps (e.g. I assume they cannot be larger than the size of the CK)? How are polar regions / edges of the data set handled? As already mentioned above, why are the climate product data smoothed only over ocean but not over land – if smoothing is necessary, it should be necessary everywhere. How are costal regions (where both land and ocean are in a 1 deg x 1 deg box) handled?

14. p. 24, Section B2:
If the asymmetry of the SCIAMACHY scan pattern is handled via the spatial merging of the SCIAMACHY pixels, why is the merging pattern symmetric (e.g. 5–6–5, see Section 3.1)? Furthermore, since SCIAMACHY is the reference for the offset correction (Section 3.3), any bias of SCIAMACHY will be in the end product (in contrast to what is written at the end of this section). Or, is there an additional offset correction for SCIAMACHY?

**Technical Corrections:**

1. Please check bracketing for citations in the text – sometimes brackets are missing or at the wrong place.

2. p. 5, l. 4:
   table → Table

3. p. 7, l. 29:
   continues → continued

4. p. 8, l. 19:
   table → Table

5. p. 23, l. 18:
   TCWV_smooth → make 'smooth' an index

6. p. 24, l. 28:
   SCIACHY → SCIAMACHY

---

## Referee Comment (RC2) · R. Lang (Referee) · 22 Nov 2017

Water vapour retrievals from the visible region (here the red part) of the spectrum contribute to the overall suite of available water vapour remote sensing retrievals by offering a very good accuracy over land and a very acceptable accuracy over ocean under day-light conditions. Like their companion retrievals from the thermal infrared region, VIS-NIR WV retrievals are sensitive to clouds, which have to be screened out, at least to a large extent. The suite of available retrieval methods for UV-VIS-NIR instruments like GOME-1 and 2, and SCIAMACHY (and also OMI) have also the advantage

to be relatively free of a priori knowledge of the atmospheric state and can therefore be considered as truly independent measurements both with respect to ground-based measurements and WV concentrations provided by model output.

The provision of a quality controlled and consistently reprocessed merged data-set from the GOME-1 and 2 and SCIAMACHY suite of instruments, already covering two decades of data and (with Metop-B and C) to be continued until the end of the 2020th, therefore provides a very important basis for validation and trend analysis.

The paper by Beirle et al. describes the strategies used to merge the three WV data-sets available using a retrieval strategy originally developed by Wagner et al and implemented in the GOME operational processor environment (GDP) by DLR (Grossi et al.). Since the retrieval method, using the 640 nm region, as well as the conversion from SCD to VCDs using AMFs derived from the oxygen B-band, are the same for all instruments, the only remaining issue is to overcome instrument design and operation specific differences and their impact on the climatological WV values derived as "cloud-free" monthly means at global coverage. Cloud screening is carried out by comparing the retrieved ground-pixel averaged mean scattering-height (derived from the O2-AMF) to the theoretical cloud-free height (using pre-scribed "maximum" pressure profiles per instrument above a cloud-free pacific-ocean), while the strategy for merging of the data, overcoming instrument specific design aspects, is to reduce the spatial resolution of GOME-2 and SCIAMACHY to that of the approximately 8 times coarser GOME-1 foot-print and subtract a global (SCIAMACHY) or latitude depended (GOME-2) offset.

The paper is well written and the results are scientifically sound so that I can recommend it for publication in ESS, provided the authors can address the two main issues I have with the presented results and analysis.

I) The reason why the offsets between SCIA and GOME-2 are so different for land and ocean is still very puzzling after having read the paper, since for me it seems

that this cannot be explained by the interference of the missing/screened out monthly narrow scan mode (NSM). This is because such instrument operations issues, like most calibration issues, should be largely independent from the surface type, as also Figure A1 - showing the impact of the NSM removal on the mean scan angle - seems to demonstrate. One can only guess that the observed land/ocean difference in the offset between the two instruments WV data might be more related to the SCD retrievals, e.g. in case they use different surface treatments or databases or similar auxiliary data related to different surface types.

II) Statistical effects from monthly averages in combination with cloud screening are not considered. This is an oversight which is often made when evaluating monthly mean data-series, especially for species with such high temporal variability. If the cloud screening per instrument, and during the course of one month, is different due to differences in spatial sampling or due to the different coverage (e.g. by the SCIAMACHY alternate limb/nadir sounding), this can lead to significant monthly differences in the averaged result, simply because the monthly average covers different days, and since the day-to-day WV variations at the same point can be very large. This may in particular play a significant role for the gaps analysis (Section 4.2), and in particular in the tropics where the temporal variability (in absolute terms) is very high. So a comparison of the WV distribution around the mean in selected months (or for each month for a selected year) should be carried out, for analysing this effect on the presented monthly mean values.

Specific comments:

Section 2.4, p.4, last paragraph: But this "consistency" could also be achieved by, e.g., using the same independent pixel approach (e.g. FRESCO+) applied to the O2-B band. And it is not clear to me why a DEM is not used for clearly separating a mountain or a cloud (using the O2-AMF approach)?

Figure 2: Why is the corresponding GOME-2 situation not shown here? I think this

would be very useful for the reader and for the user of the data-set as well.

Editorials:

Section 1, p.2, l.7: I would add "ground-based" here since GPS can also be satellite.

Section 1, p.2, l.9+10: Radio occultation (upper troposphere/stratosphere) - should be mentioned here, since RO is meanwhile a key-contributor to NWP for WV.

Section 1, p.2, l15: I would add here that therefore these measurements are "complementary to the MW, RO and IR derived climate data-sets, which are sensitive only to specific surfaces or altitude ranges".

Section 1, p.2, l19: Isn't GDP in the version 4.8, meanwhile?

Section 1, p.2, l29: I would indicate here already what the difference in the footprint is, by adding "to the smaller swath with larger ground footprint" ... or similar.

Section 1, p.2, l31: I would add that this is why this makes it also a truly independent data-set for model evaluation and evaluation of other WV climatologies.

Section 2.2, p.3, l23-24: Revise for better reading.

General: "saturation effect" should be explained somewhere, for the non-retrieval DOAS expert as being another name for the "non-linearity" effect in the spectral absorption.

Section 2.3, p.4, l5: "by" Wang et al.

Figure 4: The caption misses a description of the different SCIAMACHY data-sets displayed.

References: Danielczok, A. and Schröder, M. - The reference is not complete here, although provided in the text.

Figure A1: Caption should state that it is a "monthly" mean.

[Figure]

---

## Author Comment (AC1) · 19 Jan 2018

**Reply to Referee #2 (R. Lang)**

We would like to thank the referee Rüdiger Lang for the very constructive and helpful comments and questions.

Below we reply to the issues raised by the referee point by point, where
blue sans serif type repeats the reviewer's comments,
black serif type is used for our reply, and
*black italic type indicates text from the manuscript, whereby modifications are highlighted in red.*

Water vapour retrievals from the visible region (here the red part) of the spectrum contribute to the overall suite of available water vapour remote sensing retrievals by offering a very good accuracy over land and a very acceptable accuracy over ocean under day-light conditions. Like their companion retrievals from the thermal infrared region, VIS-NIR WV retrievals are sensitive to clouds, which have to be screened out, at least to a large extent. The suite of available retrieval methods for UV-VIS-NIR instruments like GOME-1 and 2, and SCIAMACHY (and also OMI) have also the advantage to be relatively free of a priori knowledge of the atmospheric state and can therefore be considered as truly independent measurements both with respect to ground-based measurements and WV concentrations provided by model output.
The provision of a quality controlled and consistently reprocessed merged data-set from the GOME-1 and 2 and SCIAMACHY suite of instruments, already covering two decades of data and (with Metop-B and C) to be continued until the end of the 2020th, therefore provides a very important basis for validation and trend analysis.
The paper by Beirle et al. describes the strategies used to merge the three WV datasets available using a retrieval strategy originally developed by Wagner et al and implemented in the GOME operational processor environment (GDP) by DLR (Grossi et al.). Since the retrieval method, using the 640 nm region, as well as the conversion from SCD to VCDs using AMFs derived from the oxygen B-band, are the same for all instruments, the only remaining issue is to overcome instrument design and operation specific differences and their impact on the climatological WV values derived as "cloud-free" monthly means at global coverage. Cloud screening is carried out by comparing the retrieved ground-pixel averaged mean scattering-height (derived from the O2-AMF) to the theoretical cloud-free height (using pre-scribed "maximum" pressure profiles per instrument above a cloud-free pacific-ocean), while the strategy for merging of the data, overcoming instrument specific design aspects, is to reduce the spatial resolution of GOME-2 and SCIAMACHY to that of the approximately 8 times coarser GOME-1 foot-print and subtract a global (SCIAMACHY) or latitude depended (GOME-2) offset.
The paper is well written and the results are scientifically sound so that I can recommend it for publication in ESS, provided the authors can address the two main issues I have with the presented results and analysis.

We thank the reviewer for his positive general assessment.

I) The reason why the offsets between SCIA and GOME-2 are so different for land and ocean is still very puzzling after having read the paper, since for me it seems that this cannot be explained by the interference of the missing/screened out monthly narrow scan mode (NSM). This is because such instrument operations issues, like most calibration issues, should be largely independent from the surface type, as also Figure A1 - showing the impact of the NSM removal on the mean scan angle - seems to demonstrate. One can only guess that the observed land/ocean difference in the offset between the two instruments WV data might be more related to the SCD retrievals, e.g. in case they use different surface treatments or databases or similar auxiliary data related to different surface types.

The climate product aims to create consistent time series as best as possible. This is achieved by spatial merging of SCIAMACHY and GOME-2 pixels to the coarser pixel size of GOME.
As shown in Fig. 4, the merging of SCIAMACHY pixels result in far better agreement to GOME. But still, there are small (~1-2%), but systematic differences remaining which require an additional offset

correction between the instruments. These offsets have been found to be considerably different for land and ocean, as shown in Fig. 9.

Note, however, that the retrieval settings are identical for the different instruments, i.e. the land/ocean difference cannot be explained by surface treatment or auxiliary data used within the retrieval. Instead, as discussed in section 3.3, the remaining offsets might be caused by different instrument characteristics (like polarization sensitivity or spectral resolution) or imperfect spatial re-sampling (since the SCIAMACHY pixels cannot be merged to GOME pixel size exactly).

Most of all, however, we consider the difference in local time to be the most likely cause for the remaining offsets, as we discuss in section 3.3. This is supported by the finding that the offsets GOME-SCIAMACHY and GOME2-SCIAMACHY have opposite signs, i.e. the changes over instruments are monotonous in local time. This is probably caused by a systematic change of cloud conditions between 9:30, 10:00, and 10:30 local time. Note that (a) even small changes of cloud conditions might affect the TCWV retrieval, and (b) the actual magnitude and sign of such a change is difficult to quantify, as a change in $O_2$ affects both the AMF correction and the cloud flag.

We have extended this discussion in section 3.3 of the revised manuscript to clarify the line of argumentation.

The discussion of scan angle effects and the need for a smoothed TCWV over ocean was obviously not clear enough in the manuscript. We have thus largely revised the respective section (formerly 4.4, now 3.4, as the smoothed TCWV is part of the Climate product, thus its description should be part of section 3 on "The climate product") and Appendix A on mean scan angles.

In short, (a) mean scan angles vary in monthly means, but (b) are about 0 in the long term average, except (c) in cases where the orbital patterns are not regular, such as for GOME over the Himalaya or for GOME-2 where NSM orbits are periodically missing.

The effect of such a systematic scan angle bias is now very different over land (where it has almost no effect) and over ocean (where it has a quite strong effect). This has been shown in Figure 1 of Grossi et al. (2015). Thus, orbital patterns can be recognized in the mean difference map between GOME-2 and SCIAMACHY, but only over ocean. For the offset correction, this effect is eliminated by applying just a latitude-dependent offset over ocean for GOME-2. In the final product, TCWV$_{smooth\_ocean}$ is provided, where the remaining (very small, but) systematic orbital structures over ocean are removed.

II) Statistical effects from monthly averages in combination with cloud screening are not considered. This is an oversight which is often made when evaluating monthly mean data-series, especially for species with such high temporal variability. If the cloud screening per instrument, and during the course of one month, is different due to differences in spatial sampling or due to the different coverage (e.g. by the SCIAMACHY alternate limb/nadir sounding), this can lead to significant monthly differences in the averaged result, simply because the monthly average covers different days, and since the day-to-day WV variations at the same point can be very large. This may in particular play a significant role for the gaps analysis (Section 4.2), and in particular in the tropics where the temporal variability (in absolute terms) is very high. So a comparison of the WV distribution around the mean in selected months (or for each month for a selected year) should be carried out, for analysing this effect on the presented monthly mean values.

We agree that sampling effects are important for the evaluation of the representativeness of the monthly means provided in the climate product.

Part of the effects described by the reviewer, i.e. differences between instruments caused by different spatiotemporal sampling, are shown in the comparison between GOME and SCIAMACHY for all available measurements (Fig. 3a) versus the comparison of coincident measurements only (Fig. 3b). Note that the final offset between GOME and SCIAMACHY with reduced resolution (Fig. 3c) is determined for coincident measurements during the overlap period, i.e. there are no sampling effects (except the small time shift of half an hour). For the offset between SCIAMACHY and GOME-2, on the other side, a very long overlap period of more than 5 years is available so that statistical sampling effects cancel out.

In order to provide additional information on the representativeness of the provided monthly means, we have processed a new version (v2.2) of the climate product which now also includes the monthly standard deviation (std) of TCWV for each 1° pixel and month. This reflects the day-to-day variability of TCWV within a month.

In addition, the standard error of the mean can now be derived which represents the precision of the climate product.
The standard deviation and standard error of the mean are introduced and discussed in a new section (3.5) of the revised manuscript. Additional figures of std and standard error are provided in a new Appendix D.

The systematic impact of cloud screening is a general problem of satellite remote sensing, and particularly important for water vapor. We thus add the following to section 4.3 ("Known issues: accuracy"):

*In addition, the selection of cloud free observations corresponds to generally dryer atmospheric conditions, which likely results in low biased means. This effect is unavoidable for water vapor retrievals from satellite measurements in the visible range, where clouded scenes have to be masked out.*

We agree that consistent treatment of cloud effects might also be realized by other methods, but the simple $O_2$ threshold approach does it implicitly.
We modify the respective sentence to
*The advantage of this approach  is that  directly provides a simple, but consistent  treatment of cloud effects across the different satellite instruments, ...*

And it is not clear to me why a DEM is not used for clearly separating a mountain or a cloud (using the O2-AMF approach)?

We agree that the very simple procedure (using the measured $O_2$ SCD in relation to the maximum $O_2$ SCD over the Pacific for both AMF correction and cloud masking) might generally be replaced by a more complex method which takes the surface elevation into account. This would probably improve the accuracy of the TCWV over elevated terrain and would overcome the persistent gaps over high mountains.
However, within this study, we follow the simple approach, in order to be consistent with the GDP algorithm documented in Grossi et al. (2015), and because a modified treatment of the $O_2$ threshold would require a major reprocessing of the complete data set, including updated testing and validation of the resulting product.

In the revised manuscript, we now point out at beginning of section 2 that the current retrieval includes some simplifications, which however affect all instruments likewise and thus do not impair trend analyses.

Figure 2: Why is the corresponding GOME-2 situation not shown here? I think this would be very useful for the reader and for the user of the data-set as well.

We appreciate the reviewer's comment and acted on his suggestion:
We have changed Fig. 2 by considering the 1st of June 2009 now (instead of 2003), where measurements from GOME, SCIAMACHY, and GOME-2 are available over the Atlantic.
This figure illustrates the merging patterns of small pixels into GOME-like pixels for both SCIAMACHY and GOME-2. In addition, it shows the reduction of the GOME-2 swath width, and illustrates why a direct comparison of coincident measurements between GOME-2 and SCIAMACHY (with shifted orbital patterns) is not as meaningful as between SCIAMACHY and GOME.
The reference to and discussion of Fig. 2 is revised accordingly in the manuscript.

We have modified the sentence accordingly.

Section 1, p.2, l.9+10: Radio occultation (upper troposphere/stratosphere) - should be mentioned here, since RO is meanwhile a key-contributor to NWP for WV.

We thank the reviewer for this hint and have added the following sentence to the introduction:
*In addition, radio occultation (RO) is an accurate method to determine water vapor concentrations in the upper troposphere/lower stratosphere region and is a key contributor to numerical weather prediction.*

Section 1, p.2, I15: I would add here that therefore these measurements are "complementary to the MW, RO and IR derived climate data-sets, which are sensitive only to specific surfaces or altitude ranges".

We appreciate the reviewer's proposal. We have added the following to the introduction:
*Thus, TCWV products from satellite observations in the red spectral range are a valuable complement to MW, IR and RO water vapor products, which are sensitive only to specific surfaces or altitude ranges. TCWV products derived from GOME, SCIAMACHY and GOME-2 have already been used to investigate the water vapor evolution over time on global scale, e.g. the effects of El Nino (Wagner et al., 2005; Loyola et al., 2006) or trends (Wagner et al., 2006; Mieruch et al., 2008, 2011, 2014).*

Section 1, p.2, I19: Isn't GDP in the version 4.8, meanwhile?

We have modified the respective sentence to
*The TCWV retrieval implemented in the operational GOME-2 data processor (GDP) (from version 4.7 on) ...*

Section 1, p.2, l29: I would indicate here already what the difference in the footprint is, by adding "to the smaller swath with larger ground footprint" ... or similar.

We have modified the sentence as follows:
*This consistency is reached by*
*a) spatial merging of the smaller SCIAMACHY and GOME-2 pixels (60/80 km across track) to the GOME pixel width (320 km) and*
*b) limiting the broader GOME-2 swath (1920 km) to that of GOME and SCIAMACHY (960 km).*

Section 1, p.2, l31: I would add that this is why this makes it also a truly independent data-set for model evaluation and evaluation of other WV climatologies.

We thank the reviewer for this suggestion.
We have added the sentence
*The resulting climate product is a valuable, independent data set for model evaluation, comparison to other water vapor products, and trend analyses.*

Section 2.2, p.3, l23-24: Revise for better reading.
General: "saturation effect" should be explained somewhere, for the non-retrieval DOAS expert as being another name for the "non-linearity" effect in the spectral absorption.

We have completely revised the respective section 2.2 as follows:.
***2.2 Correction of nonlinearity in spectral absorption***
*The spectrally fine structured absorption bands of water vapor are not resolved by the considered satellite instruments. Consequently, the relationship between the actual TCWV and the retrieved $H_2O$ SCD becomes nonlinear. The same holds for $O_2$.*
*This effect can be simply modelled based on synthetic spectra as described in Wagner et al. (2003, 2006) for $H_2O$ and $O_2$, respectively. For the GDP and the climate retrieval, the $H_2O$ and $O_2$ SCDs resulting from the DOAS analysis are corrected accordingly for nonlinearities in spectral absorption. This correction is also denoted as "saturation correction" in Wagner et al. (2003).*

Section 2.3, p.4, l5: "by" Wang et al.

Corrected.

Figure 4: The caption misses a description of the different SCIAMACHY data-sets displayed.

We have revised the caption to Fig. 4 to
*Top: Zonal mean TCWV for GOME and SCIAMACHY (in original as well as reduced resolution) as function of latitude.*

*Bottom: Differences of zonal mean TCWV between GOME and SCIAMACHY in original (light) and reduced (dark) resolution, separately for land (orange) and ocean (blue).*

References: Danielczok, A. and Schröder, M. - The reference is not complete here, although provided in the text.

The validation report by Danielczok and Schröder is now publically available via the ESA GOME web page:
https://earth.esa.int/documents/700255/1525725/GOME_EVL_L3_ValRep_final/db7e72c3-044d-4236-9dee-d88405b89ef0
We have added the respective link to the reference.

Figure A1: Caption should state that it is a "monthly" mean.

Figure A1 actually displays the scan angles averaged over the full available time period for each instrument. This is now specified in the figure caption.

---

## Author Comment (AC2) · 19 Jan 2018

**Reply to Anonymous Referee #1**

We would like to thank referee #1 for the very constructive and helpful comments and questions.

Below we reply to the issues raised by the referee point by point, where
blue sans serife type repeats the reviewer's comments,
black serife type is used for our reply, and
*black italic type indicates text from the manuscript, whereby modifications are highlighted in red*.

**General Comments:**
The paper by Beirle et al. describes a specific water vapour data set, which is based on
a combination of existing water vapour products for GOME, SCIAMACHY and GOME-2.
Although the underlying data and methods are not new, the resulting combined and
homogenised data set may be useful for future climate studies.
The data set is available via the specified link, but only to registered users; therefore download (and
the data product itself) could not be checked.

We have contacted the World Data Climate Center (WDCC) and discussed the reviewer's concern. Indeed, a registration is required for downloading the data. However, the registration information is kept confidential by WDCC and is not accessible for the data providers (like me). The reason for WDCC asking for registration is to be able to contact the data users, e.g. in case a dataset is withdrawn or an update is available.

Note that the climate product is a deliverable of the ESA "GOME Evolution" product and its final version will also be freely available at ESA within the coming weeks:
https://earth.esa.int/web/sppa/mission-performance/esa-missions/ers-2/gome/products-and-algorithms/products-information

In addition, the climate product v2.2 is available at

ftp://ftp.mpic.de/GOME_Evolution/climate/v2.2/

The paper is well written and contains (except for the points listed below) all required
information for a potential data users, including information about the quality of the data set.

We thank the reviewer for his/her general assessment.

However, it is not clear how some of the references especially in the validation
section can be accessed (see below).

The validation reports are now publically available via the ESA GOME web page:

Danielczok and Schröder (2017):
https://earth.esa.int/documents/700255/1525725/GOME_EVL_L3_ValRep_final/db7e72c3-044d-4236-9dee-d88405b89ef0

Grossi (2017):

https://earth.esa.int/documents/700255/1525725/Grossi_GOME-Evo_Comparison_results_timeseries/9809b5b0-8f5d-4dc5-8824-dbdcaddc5174

We have added the respective links to the reference list.

My main points of criticism are:

1. Scan angle correction:
It is stated in Section 2.3 that this is not applied for the climate data product because
of the smaller scan angle range and the complexity/quality of the correction.
However, as shown in Fig. A1, there are especially for GOME-2 significant
scan angle effects.

We thank the reviewer for the detailed inquiry of the scan angle correction here and below. Obviously, we have to improve our discussion and line of arguments here.

First of all, we would like to point out that while there are systematic biases in the monthly and even the overall mean scan angles, the impact on the monthly/total mean TCWV is generally quite small. For instance, in September 2015 (as shown in Fig. 10), the difference between TCWV (where faint orbital patterns can be imagined) and $TCWV_{smooth}$ (now denoted more precisely as $TCWV_{smooth\_ocean}$) is about $0.0\pm1.2$ kg/m$^2$ (mean±std) over ocean (excluding coastal regions). The corresponding relative differences (TCWV- $TCWV_{smooth}$)/$TCWV_{smooth}$ are $0.00\pm0.05$ (mean±std), i.e. typically within 5%.

For the mean of all months, the respective absolute and relative differences are as low as $0.0\pm0.1$ kg/m$^2$ and $0.00\pm0.01$, i.e. within 1%.

Over land, the scan angle biases have almost no effect on TCWV (compare Fig. 1 in Grossi et al., 2015).

The reason for yet discussing the scan angle effect that intensively in the manuscript and for introducing the additional product $TCWV_{smooth\_ocean}$, though the effect is so small, is that the systematic pattern can actually still create artificial orbital patterns in trend analyses.

In the revised version of the manuscript, we have clarified the line of argumentation when discussing scan angle biases, their small but systematic impact on TCWV (over ocean), and the need for the additional data field $TCWV_{smooth\_ocean}$.

Note that we have shifted the respective section "4.4 Scan angle dependency" into a section "3.4 Smoothed TCWV over ocean", as this is a central part of the product and should thus be described within section 3 (The climate product). Within section 3.4 we motivate the need for a smoothed product, while the detailed line of argument for systematic scan angle biases and their impact are provided in Appendix B, which is slightly extended for clarification.

Within the retrieval of the climate product, we have decided to not apply an explicit scan angle correction for individual satellite pixels in order to keep the algorithm as simple as possible. In addition, the scan angle dependencies are not only considerably different over land and over ocean (see Grossi et al., 2015, Figure 1), but also depend on latitude/SZA etc. Most of all, the effective scan angle dependency of the final TCWV results from scan angle dependencies of both $H_2O$ and $O_2$ SCDs. The latter affect both the $H_2O$ AMF and the flagging of clouded pixels, which makes an appropriate correction of scan angle effects quite complex. We have modified section 2.3 accordingly in the revised manuscript.

Although the scan angle related patterns are (at least partly) removed for the offset correction, they are still left in the climate product (at least in the unsmoothed one). The product contains a corresponding warning flag, but from Fig. A1 it seems that a lot of data will be affected. The impact of not applying a scan angle correction on the data product should be quantified and a clear recommendation should be given to the data users, if the flagged data should be used or not (or, under which conditions).

Within the description of $TCWV_{smooth\_ocean}$ (now section 3.4), we have added a comparison to the unsmoothed TCWV and thereby quantify the effect of the remaining scan angle effects. They are about $0.0\pm1.2$ kg/m$^2$ (mean±std) for individual months and $0.0\pm0.1$ kg/m$^2$ for the temporal mean TCWV over ocean (excluding coastal regions).

Over land, scan angle effects are far weaker and are thus negligible, except for the region around the calibration gap for GOME, where locally only measurements of the western or eastern pixel are available. This region is indicated by the warning flag, as specified in the new Appendix C of the revised manuscript, and should generally be skipped.

We have also added a recommendation to section 3.4:

*We generally recommend to use the $TCWV_{smooth\_ocean}$ product for trend analysis.*
*For validation of the climate product or comparisons to other data products, we recommend to use $TCWV_{smooth\_ocean}$ as well, except for coastal regions where biases due to edge effects of the convolution with $C_{smooth}$ have to be expected (note that this effect does not affect trend analyses). Here, TCWV should be used. The potentially affected coastal regions are specified by a "convolution flag" which is also provided in the data product and explained in Appendix C.*

2. Product errors:
The climate data product does not contain any TCVW errors. There is no description on how errors of the underlying products are considered. If it is not

possible to specify an error for each data point, at least some general information about the expected quality (independent from validation results) should be given. For example, I would expect different quality/errors for the different instruments (and therefore for different time intervals), simply because of the merging of SCIAMACHY and GOME-2 pixels to GOME spatial resolution.

The accuracy of the climate product is discussed in section 5.1 based on validation results. In response to the reviewers' demand for an error for each data point, we now provide an updated version of the climate product which also includes the standard deviation (in addition to the monthly mean TCWV) and the number of available days for each $1°x1°$ pixel for each month.
The standard deviation reflects the day-to-day variability of TCWV during one month.
In addition, the standard error of the mean can now be derived which represents the precision of the climate product. While the std is similar for the different instruments, the standard error (std/sqrt(n)) is highest for SCIAMACHY alone (due to measurement gaps during limb mode) and lowest during overlap periods.
The standard deviation and standard error of the mean are introduced and discussed in a new section (3.5) of the revised manuscript.

**Specific Comments:**

1. p. 4, l. 7–10:
Using the $O_2$ AMF as proxy for $H_2O$ AMF assumes that the $O_2$ VCD is known. What is usually known is the $O_2$ VMR, but the VCD should also depend on (varying) pressure and temperature. The additional correction factor for different profile shapes is only determined from standard atmosphere conditions. Is there a remaining dependency on pressure and temperature? Since this applies to both $O_2$ and $H_2O$ in a similar way the final impact might be low, but maybe this should be mentioned in the text.

Within the GDP retrieval as well as within the climate product, variations of the $O_2$ VCD are neglected, as they are rather small (about 5%). Note that (a) other (systematic) effects, in particular the correlation between pressure systems and cloud conditions, or systematic deviations of the vertical water vapor profile from the apriori, have typically a much stronger impact on the retrieval, and (b) this affects all instruments likewise, thus has no impact on trends.

In the revised manuscript, we have added
*Note that temporal variations of the actual $O_2$ VCD due to pressure and temperature variations are neglected, as they are far smaller than other potentially systematic effects like cloud conditions depending on pressure systems.*

2. p. 4, l. 22–23:
Where does the range 2–3 km come from – is this related to the 80% $O_2$ SCD threshold? I assume this should depend also on cloud fraction?

Yes, the 80% threshold for $O_2$ corresponds to an altitude of about 2 km. The detailed radiative transfer effects depend on cloud fraction, surface albedo etc., but here we just want to roughly point out two general implications of the simplified approach.

We have revised the respective paragraph to
*At altitudes above 2 km, pressure is reduced to less than 80%. Consequently, mountains above about this altitude (on GOME horizontal resolution) are generally skipped by the simple $O_2$ cloud masking, while clouds below this altitude are kept.*

3. p. 5, l. 30–31:
The along track spatial resolution is given by the product of the along-track velocity of the satellite and the measurement (integration) time (plus along track size of the field of view). Since orbital parameters and along track field of view sizes of GOME and SCIAMACHY are very similar, the main difference is the integration time. Merging the SCIAMACHY data such that the across track spatial resolution matches the one of GOME should therefore result in a quite similar along track spatial resolution. So, actually the merging accounts for the difference in along track extent, which is then even smaller than the mentioned 30 km vs. 40 km.

The orbital parameters of GOME and SCIAMACHY are indeed very similar, as also illustrated in Fig. 2. Also the integration times (IT) for a complete forward scan (without backscan) are quite similar: for GOME, IT is 3x1.5 s = 4.5 s (see GOME manual, page 41); for SCIAMACHY typical pixel size (60 across track), IT is 16x0.25 s = 4s (see SCIAMACHY book, figure 4.9, page 57). The along-track instantaneous field of view, however, is significantly smaller for SCIAMACHY (1.8°, see SCIAMACHY book, page 35) than for GOME (about 2.7°, see GOME manual, page 57), i.e. SCIAMACHY really does provide better spatial resolution in the along-track dimension, and thus also cloud statistics have to be expected to be different for GOME and "reduced" SCIAMACHY pixels. This effect, however, cannot be resolved by a simple merging scheme of SCIAMACHY ground pixels.

References:

GOME manual: https://earth.esa.int/documents/10174/1596664/GOME05.pdf

SCIAMACHY book: Gottwald, M. and Bovensmann, H.: SCIAMACHY - Exploring the Changing Earth's Atmosphere, 1st Edition., Springer Netherlands, ISBN 90-481-9895-X, 2010.

**4. p. 7, l. 10:**
Is there a reason for the larger differences in the tropics? Could this be related to the above mentioned issue that corrections for the different profile shapes are based on average atmospheric conditions or maybe the probably in the tropics larger (and possibly less accurate) saturation correction? Please discuss.

The main reason for the larger difference in the tropics is that the TCWV itself is highest in the tropics. Latitudinal dependency of relative differences (with respect to the TCWV mean) are less pronounced. Note that a potential bias due to the simplified assumption of a constant $O_2$ profile would affect all instruments likewise and cannot explain the observed inter-satellite discrepancies.
The saturation correction was determined in Wagner et al. (2003) for the GOME instrument. We have checked the impact of the slightly different spectral properties of SCIAMACHY and GOME-2 on the saturation correction, and found low effects (<1%) for both $H_2O$ and $O_2$ SCDs at low and mid-latitudes. Since these systematic effects mostly cancel out by the application of the $O_2$ AMF, they can be neglected. Only at high latitudes (for high SZA), the impact on the $O_2$ SCD can be up to 3%. The respective effect on $H_2O$ introduced by the $O_2$ AMF is very low in absolute TCWV and corrected by the applied offset correction. This is discussed in section 2.2 of the revised manuscript.
We consider the difference in local time to be the most likely cause for the observed differences over the tropics. This is supported by the fact that the offset between GOME and SCIAMACHY is almost mirrored to the offset between GOME-2 and SCIAMACHY. I.e., there is a monotonous change from 9:30 via 10:00 to 10:30 local time. Such a systematic effect of the retrieved TCWV with local time can easily be caused by a change of cloud conditions, which would affect both $O_2$ AMF and cloud masking.
We have extended this line of argument in section 3.3.

**5. p. 10, l. 24–26:**
Why is the smoothing only applied to ocean data? The scan angle dependent artefacts also occur over land (see Fig. A1).

The mean scan angle biases are comparable over ocean and land, as shown in Fig. A1, but their impact on the TCWV retrieval is much stronger over ocean than over land, as shown in Figure 1 in Grossi et al. (2015).
We have extended the discussion of the impact of scan angle biases in the respective section (formerly 4.4, now 3.4) for clarification.

**6. Section 5:**
The section on validation mainly refers to a report by Danielczok and Schröder (2017). Where is this report available?

The report by Danielczok and Schröder is now publically available via the ESA GOME web page. We have added the respective link to the reference.

If it is not available to the public a bit more information about the validation activities should be given (collocation criteria, averaging procedures etc.).
Furthermore, from the description it is not clear where the GNSS/ARSA stations

are located – I suggest to add a corresponding map. Probably these stations
are mostly at land, so how good is the information about the quality of the data
product over ocean?

Maps of the distribution of contributing GNSS and ARSA stations are provided in Danielczok and Schröder (2017):
https://earth.esa.int/documents/700255/1525725/GOME_EVL_L3_ValRep_final/db7e72c3-044d-4236-9dee-d88405b89ef0
The reviewer is right in stating that these stations are mostly located over land, and thus the validation is basically restricted to land, with a particular focus on North America and Europe.

The quality of the GDP data product over ocean has been investigated by Grossi et al. (2015). Similar comparisons for the climate product have been performed in Grossi (2017) (see next point).

We have extended the discussion in section 5 (validation) by clearly pointing out the restriction of the comparison to GNSS and ARSA to land and discussing the quality of the climate product over ocean based on comparisons to ECMWF and SSM/I HOAPS4.

7. p. 11 , l. 18:
Where is the reference by Grossi(2017) available?

The report by Grossi is now publically available via the ESA GOME web page:
https://earth.esa.int/documents/700255/1525725/Grossi_GOME-Evo_Comparison_results_timeseries/9809b5b0-8f5d-4dc5-8824-dbdcaddc5174
We have added the respective link to the reference.

8. p. 12 , l. 10–11:
'Note that the transitions between the satellite instruments are not visible in the
timeseries of the difference to the validation data sets.'
Is this because of the averaging of data during the overlap times?

The aim of the applied offset corrections is to create consistent time series of TCWV across different satellite instruments without "jumps". The statement was meant to confirm that this actually worked out.

We have modified the sentence to
*Note that in the comparison to the different validation data sets, the transitions between the different satellite instruments can not be identified any more, i.e. the application of the offset correction succeeded in creating a long-term consistent time series of TCWV.*

9. p. 12, l. 15:
It should be mentioned earlier (e.g. in the introduction) that the climate product
contains monthly mean data at 1 deg resolution.

We agree. The introduction is thus modified as follows:
*Within the ESA GOME-Evolution project,.the "Climate"  product has been developed, which provides monthly mean TCWV from July 1995 to December 2015 at 1° resolution.*

10. p. 12, l. 28:
How many data are affected by the Warning_Flag? Looking at Fig. A1 and noting
that there is no distinction between time and/or instrument this should be quite
a lot. I suggest to add a plot of the Warning_Flag to Fig. A1 and an average
number of affected data points to have a better impression on this.

We have extended the warning flags provided in the climate product. They are described (including figures and the number of affected pixels) in a new section C of the Appendix.

11. Fig. 2:
It seems that the GOME ground pixels across track have different size (centre
pixel is smaller) whereas coadded SCIAMACHY pixels have about the same size.
Could this have been improved by different coadding patterns for SCIAMACHY?

Unfortunately, there is no perfect match possible between SCIAMACHY and GOME-1. We have decided for a co-adding pattern of the 16 SCIAMACHY forescan pixels into groups of 5/6/5 pixels mainly for

symmetry reasons. By this choice, the center SCIAMACHY pixel is indeed larger than the east&west pixels, whereas for GOME, the center pixel is slightly smaller.

However, any alternative pattern has some other drawbacks:
- the only symmetric pattern with a smaller center pixel would be 6/4/6; for this case, however, the center pixel would have a width of 240 km, which would be far smaller than the GOME extent.
- any other option would lose symmetry (5/5/6 or 6/5/5). For GOME-2, however, the merging scheme 24->8/8/8 is obvious and symmetric. Concerning the issue of scan angle dependencies, we consider it important to keep symmetry of the merging patterns for all instruments. Thus, we consider the choice of the pattern 16->5/6/5 for SCIAMACHY to be the best compromise.

**12. Figs. 13 & 14:**
Fig. 13 shows a small decrease of the differences towards the end of the time series (probably the reason for the negative stability). This decrease is not seen in Fig. 14, which indicates that it is not related to the satellite data (e.g. due to GOME-2 degradation). Is there any information about the quality of the station data?

We have extended the discussion of Fig. 13 in the revised manuscript:

*Note that the decrease of differences between the climate product and GNSS TCWV at the end of the time series (Fig. 13) does not appear in similar comparisons to ARSA (Fig. 14). In addition, it also does not appear in the GNSS intercomparison if only coincident measurements are considered (see Fig. 3-8 in Danielczok and Schröder. 2017). The different results are probably caused by the different spatial distribution of stations, and sampling effects of the Climate product.*

**13. Appendix A:**
A bit more information should be given about the 'normalised convolution' and how it is exactly applied.

We have extended appendix A for clarification.

What are the limitations for removing gaps (e.g. I assume they cannot be larger than the size of the CK)?

We have modified the respective sentence as

*normalized convolution can be applied to matrices containing gaps and removes them (as long as the extent of the CK is larger than the gap)*

How are polar regions/edges of the data set handled?

We have added the following information to the manuscript:

*For convolution, the grid is considered to be cyclic in longitude (i.e. smoothing across the dateline is done appropriately), but finite in latitude (i.e., no smoothing is applied across the poles).*

As already mentioned above, why are the climate product data smoothed only over ocean but not over land – if smoothing is necessary, it should be necessary everywhere.

The mean scan angle biases are comparable over ocean and land, as shown in Fig. A1, but their impact on the TCWV retrieval is much stronger over ocean than over land, as shown in Figure 1 in Grossi et al. (2015).
We have extended the discussion of the impact of scan angle biases in the respective section (formerly 4.4, now 3.4) for clarification.

How are costal regions (where both land and ocean are in a 1 deg x 1 deg box) handled?

We have applied a simple flag for categorizing land and ocean by scaling down GTOPO30 on 1° resolution. A 1°x1° pixel is categorized as oceanic if more than 35% of the respective GTOPO30 pixels are ocean pixels.
The applied ocean mask is provided as auxiliary data to the updated TCWV product.

14. p. 24, Section B2:
If the asymmetry of the SCIAMACHY scan pattern is handled via the spatial
merging of the SCIAMACHY pixels, why is the merging pattern symmetric (e.g.
5–6–5, see Section 3.1)?

The main intentions of the merging procedure are to create pixels of
a) as best as possible comparable size, in order to allow for consistent treatment of cloud effects, and
b) preserve symmetry between the size of the Eastern and Western pixel (which is given for GOME and GOME-2) also for SCIAMACHY.
The slight asymmetry of the SCIAMACHY scan pattern of about 1° corresponds to an across-track distance of about half a SCIAMACHY ground pixel, thus it cannot be accounted for by a different merging pattern.

Furthermore, since SCIAMACHY is the reference for
the offset correction (Section 3.3), any bias of SCIAMACHY will be in the end
product (in contrast to what is written at the end of this section). Or, is there an
additional offset correction for SCIAMACHY?

The reviewer is right that a possible absolute bias of SCIAMACHY will not be removed by the offset correction; only the relative biases between the instruments are corrected.
Note however that a potential systematic bias caused by the systematic small asymmetry of the SCIAMACHY scan pattern would cause offsets of the same sign when compared to GOME or GOME-2. We however found offsets of opposite signs, i.e. there are far stronger effects causing the observed offsets, most likely the change of cloud conditions with local time.
In the revised manuscript, we have revised the last sentence of section B2 as follows:
*Consequently, any (small) bias between the different instruments potentially caused by the systematic negative SCIAMACHY scan angles is corrected for by contained in the offsets determined during overlap periods.*

**Technical Corrections:**

We have checked and corrected the bracketing of citations throughout the document.

table → Table
Corrected.

continues → continued
Corrected.

table → Table
Corrected.

TCWV_smooth → make 'smooth' an index
We now denote the smoothed TCWV as $TCWV_{smooth\_ocean}$ except in section 6 where the NetCDF variable names are given, which is now `TCWV_smooth_ocean`. In addition, we introduced the symbols $V$ and $V'$ for TCWV as $TCWV_{smooth\_ocean}$ for better readability.

SCIACHY → SCIAMACHY
Corrected.